# Spatially variable soil water repellency enhances soil respiration rates (CO$_2$ efflux)

Emilia Urbanek[1] and Stefan H. Doerr[1]

[1]Department of Geography, College of Science, Swansea University, Swansea, SA2 8PP, UK

*Correspondence to*: Emilia Urbanek (e.urbanek@swansea.ac.uk)

**Abstract.** Soil CO$_2$ emissions are strongly dependent on water distribution in soil pores, which in turn can be affected by soil water repellency (SWR; hydrophobicity). SWR restricts infiltration and movement of water, affecting soil hydrology as well as biological and chemical processes. Effects of SWR on soil carbon dynamics and specifically on soil respiration (CO$_2$ efflux) have been studied in a few laboratory experiments but they remain poorly understood. Existing studies suggest that

soil respiration is reduced in water repellent soils, but the responses of soil CO$_2$ efflux to varying water distribution created by SWR are not yet known.

Here we report on the first field-based study that tests whether soil water repellency indeed reduces soil respiration, based on *in situ* field measurements carried out over three consecutive years at a grassland and pine forest site under the humid temperate climate of the UK.

CO$_2$ efflux was reduced on occasions when soil exhibited consistently high SWR and low soil moisture following long dry spells. However, the highest respiration rates occurred not when SWR was absent, but when SWR, and thus soil moisture, was spatially patchy, a pattern observed for the majority of the measurement period. This somewhat surprising phenomenon can be explained by SWR-induced preferential flow, directing water and nutrients to microorganisms decomposing organic matter concentrated in 'hot spots' near preferential flow paths. Water repellent zones provide air-filled pathways through the

soil, which facilitate soil-atmosphere O$_2$ and CO$_2$ exchanges. This study demonstrates that SWR have contrasting effects on CO$_2$ fluxes and, when spatially-variable, can enhance CO$_2$ efflux. Spatial variability in SWR and associated soil moisture distribution needs to be considered when evaluating the effects of SWR on soil carbon dynamics under current and predicted future climatic conditions.

## 1 Introduction

Soil is the most important reservoir of terrestrial carbon (C), storing four times more C than plant biomass (Karhu et al., 2014), but large amounts of C are released back to atmosphere mainly as carbon dioxide ($CO_2$) formed by microbial decomposition of organic matter as well as biological activity of roots and microfauna (Bond-Lamberty and Thomson, 2010;

Rey, 2015). Soil moisture is one of the most important environmental factors regulating the production and transport of $CO_2$ in terrestrial ecosystems (Maier et al., 2011; Moyano et al., 2012). It influences not only soil organic carbon (SOC) bioavailability and regulates access to oxygen ($O_2$) (Moyano et al., 2012; Yan et al., 2016), but also controls pore-water connectivity and therefore SOC mass transport (Davidson et al., 2012).

Soil C models consider changes in soil moisture conditions, but they use functions that represent an average response of soil

respiration to soil moisture content and do not account for within-soil moisture variability, which is a characteristic of most soils (Yan et al., 2016; Rodrigo et al., 1997; Moyano et al., 2013). Soils are typically very heterogeneous, with moisture distribution and water movement being variable and dependent on a number of factors (e.g. texture, structure, organic matter content) that determine soil hydrological properties. Soils prone to development of soil water repellency (SWR) are particularly susceptible to spatially highly variable soil moisture distribution and irregular wetting (Dekker and Ritsema,

1995; Doerr et al., 2000; Ritsema and Dekker, 2000). SWR is a common feature of many soils worldwide, and is expected to become even more widespread and severe under a warming climate (Goebel et al., 2011). SWR affects soil-water relations by restricting infiltration, which results in large areas of soil remaining dry for long periods even after substantial rainfall events (Keizer et al., 2007). It often leads to enhanced preferential flow where water moves along pathways offered not only by cracks, root channels and other types of macropores, but also zones of less repellent soil, leaving other areas completely

dry for long periods (Urbanek et al., 2015).

Preferential flow in water repellent soil is often described as fingered flow where distinct zones of vertical flow can be observed next to dry regions reaching down to subsurface soil areas (Dekker and Ritsema, 2000; Wallach and Jortzick, 2008; Urbanek and Shakesby, 2009). Such a division of soil compartments into regions of preferential water flow can create zones of elevated biological activity and organisation into so-called 'hot spots' around the water flow channels where it is easier

for microorganisms to access $O_2$, water and nutrients (Jasinska et al., 2006; Or et al., 2007; Morales et al., 2010).

Several studies have investigated microbial activity in water-repellent soils, mainly to determine whether the microbial exudates and proteins can cause the development of hydrophobic particle surfaces in soils (White et al., 2000; Feeney et al., 2006; Lozano et al., 2014). SWR has also been reported as an important factor in reducing soil microbial activity and it has been considered as one of the factors protecting soil organic C from microbial decomposition by separation of the

microorganisms from their food and water source (Piccolo and Mbagwu, 1999; Piccolo et al., 1999; Bachmann et al., 2008). Goebel et al. (2007) demonstrated that SWR affects the distribution and continuity of the liquid phase in the soil matrix and therefore restricts the accessibility of SOM and the availability of water, $O_2$ and nutrients to the microorganisms. Using laboratory-based studies, they observed lower respiration rates from soils in a water repellent state and decreasing $CO_2$ flux with increasing severity of water repellency (Goebel et al., 2005; Goebel et al., 2007). In a review of this topic Goebel et al.

(2011) highlighted the importance of SWR in organic matter decomposition especially during extreme climatic events such as drought, suggesting that it reduces the total soil $CO_2$ flux. After inducing experimental droughts, Muhr et al. (2010, 2008) speculated that a slow regeneration of $CO_2$ fluxes observed following wetting could have been caused by SWR, however, they did not actually test for water repellency. The small number of existing laboratory-based studies suggest reduced soil respiration (i.e. $CO_2$ efflux) when soil is water repellent, but a thorough field study investigating spatio-temporal changes in

water repellency and their effect on soil $CO_2$ efflux, however, is still lacking.

The aim of the current study is, therefore, to investigate, for the first time, soil $CO_2$ flux response to SWR under undisturbed *in-situ* conditions in the field. We test the hypothesis that the presence of water repellency reduces soil respiration also under 'real world' field conditions. The study sites selected were humid-temperate grassland and pine forest in the UK, which were anticipated to exhibit substantial temporal and spatial variability in SWR (Doerr et al., 2006), which is a common feature of

water repellent soils in general (Doerr et al., 2000).





## 2 Materials and methods

### 2.1 Experimental design

A forest and a grassland site, both subject to humid-temperate conditions, were chosen because of their likely high susceptibility to develop seasonal SWR in view of their sandy texture and permanent vegetation cover, which are

characteristics known to be conducive to SWR development (Doerr et al., 2000). Both study sites consisted of six plots with adjacent grass and bracken cover, arranged along a 20-m transect (Fig. 1). The sites were monitored during the growing seasons in three consecutive years (2013-2015), involving continuous measurement of soil moisture and soil temperature, and recording of $CO_2$ fluxes and persistence of SWR during site visits at approximately monthly intervals. For each study site twelve PVC collars for $CO_2$ measurements were installed, and for each vegetation plot the vegetation inside of one collar

was left intact and other had vegetation and litter layer temporarily removed for the duration of the $CO_2$ flux measurement to assess the contribution of different layers to total soil respiration.

Given the near-impossibility of finding wettable and water-repellent soils for comparison that otherwise display identical properties (e.g. texture, organic matter content, pH, litter type), we examined sites that displayed temporally variable behaviour, switching between water-repellent and wettable states of soil. This facilitated examining the impact of water-

repellency on $CO_2$ fluxes, bearing in mind that temperature and moisture themselves are known to affect SWR and $CO_2$ fluxes. C and N contents as well as pH were determined on soil samples in the laboratory to be considered as potential factors for $CO_2$ efflux variability between plots and study sites.

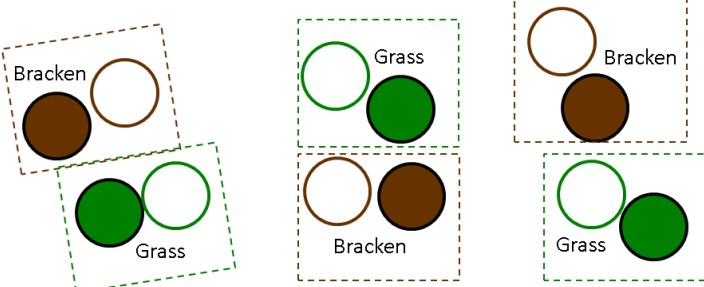

**Figure 1:** Schematic presentation of plots and $CO_2$ flux measurement collars at both, the forest (T-f) and grassland (T-g) study site. The
dashed squares identify study plots (6) and circles - soil collars for $CO_2$ flux measurements (12), green coloured shapes represent soil surface vegetated with grass and brown – with bracken; closed circles represent vegetated area, open circles – bare soil with vegetation temporarily removed.



## 2.2 Study sites

The study sites are located in eastern England, approximately 8 km north-west (grassland site (T-g); 52°24'56.42"N 0°52'31.19"E) and 8 km east (forest site (T-f); 52°27'30.82"N 0°40'50.31"E) of Thetford. The sites are subject to humid-temperate conditions with an annual mean rainfall of 665 mm spread relatively uniformly throughout the year and an annual

mean temperature of 14.5°C, with monthly mean maxima of 23°C in July and August and minima of 9°C in December and January (UK Met Office, 2017a). The site T-f is part of a long-term forest monitoring network established since 1995 aimed to assess the impact of the changing environment on forest and soil health (Vanguelova et al., 2010; Waldner et al., 2014; Jonard et al., 2015). Both sites have been planted with similar tree species, which were Scots Pine (88%), beech (6%) and oak (6%) (T-g in 1928 and T-f in 1967), but all trees at T-g were felled in 1999 and the site converted to a managed

grassland. The dominant soil cover species at both sites are essentially the same with large areas covered by either grasses (*Holcus lanatus, Agrostis canina*) or bracken (*Pteridium aquilinum, Dryopteris dilatata*). At the site T-f, however, some moss (*Eurhynchium praelongum, Rhytidiadelphus sp.*) is also present at the soil surface (UK Forest Research, 2017a). The site T-f is subject to minimal management, a few trees having been removed during the winter/spring of 2014 near the monitoring site. At the site T-g, grass mowing is conducted twice a year to control tree seedling growth. The soil type at both

study sites is Ferralic Arenosol with an approximately 3-cm thick litter layer at the T-f site, and 0-13 cm thick Ah horizon of organic rich sand with woody roots and occasional flints (UK Forest Research, 2017b). More information about the basic properties of the soils at the study sites is given in Table 1.



**Table 1:** Selected soil properties for samples (n=12) retrieved from the $CO_2$ flux monitoring collars after the field campaign had been completed. See main text for further details.

| Site | Soil depth (cm) | C content (%, mean (st.dev)) | | C:N (mean (st.dev)) | | pH (-) | | Bulk density (g cm$^{-3}$) | |
|---|---|---|---|---|---|---|---|---|---|
| | | Bracken | Grass | Bracken | Grass | Bracken | Grass | Bracken | Grass |
| T-f | 0-2.2 | 26.9 (12.1) | 7.2 (6.1) | 23.5 (2.0) | 13.2 (6.3) | 3.6 | 4.6 | 0.3 | 0.9 |
| | 2.2-4.5 | 8.3 (4.7) | 2.4 (1.5) | 16.3 (9.3) | 9.7 (6.1) | 3.7 | 5.2 | 0.7 | 1.2 |
| | 4.5-6.7 | 3.0 (2.4) | 1.5 (0.8) | 10.3 (2.7) | 7.0 (3.4) | 4.0 | 5.1 | 1.1 | 1.1 |
| | 6.7-9.2 | 1.2 (0.7) | 1.6 (0.7) | 6.6 (4.6) | 7.2 (2.6) | 4.1 | 5.2 | 1.3 | 1.3 |
| T-g | 0-2.2 | 24.3 (6.1) | 20.0 (5.3) | 23.1 (6.6) | 20.4 (8.6) | 2.9 | 3.1 | 0.5 | 0.7 |
| | 2.2-4.5 | 8.7 (4.4) | 7.4 (5.2) | 13.2 (8.3) | 12.2 (6.0) | 3.0 | 3.0 | 1.1 | 1.2 |
| | 4.5-6.7 | 3.3 (1.3) | 3.0 (2.1) | 10.5 (4.5) | 7.9 (8.0) | 3.0 | 3.1 | 1.2 | 1.3 |
| | 6.7-9.2 | 0.8 (0.1) | 1.2 (0.2) | 4.9 (1.7) | 5.7 (2.8) | 3.2 | 3.1 | 1.5 | 1.8 |



### 2.3 *In situ* monitoring of soil $CO_2$ fluxes, soil moisture and temperature

PVC collars (twelve per study site; Fig. 1) were inserted into the soil to enable $CO_2$ flux measurements to be made. The collars (20 cm diameter, 6 cm height) were inserted to a depth of 4 cm leaving the remaining 2 cm protruding above the surface. This minimal insertion depth (Heinemeyer et al., 2011) ensured that the collars remained in place allowing a sealed

contact with the chamber during the measurement, but minimised the unnatural isolation of soil and plant roots inside the collars from areas outside. For each study plot, the vegetation and the litter layers within one soil collar was temporarily removed for the duration of the $CO_2$ flux measurements and carefully put back after to avoid increased soil evaporation, while vegetation in the other collar was left undisturbed.

$CO_2$ fluxes were measured using a Li 8100A Infrared Gas Analyser (IRGA) system with a 20-cm diameter dark chamber

(LiCor Inc, Lincoln, NE, USA) placed over the installed PVC collars for the time of the measurement. The change in $CO_2$ concentration in the chamber was monitored over 2 minutes starting at the ambient $CO_2$ concentration and repeated twice for each collar at 2-minute intervals. The $CO_2$ flux was calculated based on the exponential fit of change in dry $CO_2$ concentration through time, excluding a 30-s initial phase at the start of the measurement. The results with exponential fit $R^2 < 0.95$ were not included.

During each $CO_2$ flux measurement, volumetric soil water content (SWC) was recorded with a Theta-Probe (ML3, Delta-T Devices) inserted at the soil surface up to 5cm depth next to PVC collar. Continuous monitoring of soil moisture and temperature at 5 and 10 cm depths at study plots was also conducted using soil sensors (5TM, Decagon Devices, Inc.) connected to a datalogger. During each field visit, intact soil samples were collected from each plot approx. 10 cm from the $CO_2$ flux collars using PVC tubes (5 cm diameter, 9 cm height) to allow further soil measurements under controlled

laboratory conditions. In addition to this regular soil sampling, intact soil samples from within collars were also collected at the end of the measuring campaign to determine soil properties within the collar.

Meteorological data were obtained from the Santon Downham meteorological station located 500 m from the site T-g, while a dedicated rain gauge for monitoring of precipitation was installed at the T-f site.

## 2.4 Soil sample analysis

Soil samples collected during each field visit were kept sealed in a constant temperature room for 24 hrs, then split into 4 depths (0-2.2, 2.2-4.5, 4.5-6.7, 6-7-9.2 cm) to determine their bulk density (dB), SWR and SWC. Wettability of soil was determined under field moist conditions using the WDPT test by applying 5 water drops (15 µl each) of tap water to the soil

surface of each sample and recording the time until their full infiltration (Doerr, 1998). The median value of 5 drops for each sample was used to determine the wettability "persistence" class (Doerr, 1998), wettable (<5 s), slight- (6-60 s), moderate- (61-600 s), strong- (601-3600 s) and extreme- (>3600 s) water repellency. The results were calculated and presented as WDPT frequency distribution (based on results from 6 plots & 4 depths). In addition, for determining the response of $CO_2$ fluxes to SWR conditions, the results were grouped into the SWR distribution based on the proportion of samples falling into

the extreme water repellency class per measuring event. WDPT class divisions are essentially arbitrary, but the division chosen here is based on the reasoning that presence of soil with the highest level of water repellency (i.e. extreme) has the most severe effect in terms of inducing preferential flow and thus soil water distribution.

Water content of soil samples was determined gravimetrically by drying them at 105°C for 24 hrs and converting the weights into volumetric equivalents by incorporating soil bulk density values.

Total C and nitrogen (N) contents in the soil samples were determined using a PDZ Europa ANCA GSL Elemental Analyser coupled with a 20/20 isotope ratio mass spectrometer. Samples of dried, homogenised soil were weighed in tin foil capsules and combusted over chromium oxide in helium with excess $O_2$ at 1000°C. The resulting gases were reacted over hot copper (600°C) to reduce oxides of N, $CO_2$ and $N_2$ were determined using the gas chromatography. Elemental composition and C:N ratios were calculated based upon peak areas relative to the standard reference materials acetanilide and atropine. Soil pH

was determined after 1:5 dilutions in distilled water and measured with the pH electrode.

## 2.5 Data analysis

Statistical analyses of data were performed using SPSS 22. For purpose of some data analyses the results of soil water content, soil temperature and soil water repellency distribution have been grouped into bands representing a narrow range of ± 2°C each (e.g. soil temperature within the 8°C band included values of 6.1-10 °C). Data were tested for normal distribution

and homogeneity of variance, and data with non-normal distribution and/or unequal variances were transformed (square root,

log) in order to carry out parametric analyses. A general linear model (linear mixed model) was used to identify key factors

analysed that might be affecting soil $CO_2$ fluxes using a grouped results approach. For multiple comparisons, the ANOVA

test was used to analyse significant differences. Significance of all test outcomes was accepted at p levels $<0.05$.

## 3 Results

### 3.1 Meteorological and soil conditions

The average annual temperatures and precipitation during the three years of the field monitoring campaign were very close

to 30-year average (1961 to 1990; UK Met Office, 2017b). The average air temperatures between three years of monitoring

were also similar but the precipitation patterns showed important variations (Fig. 2). Contrasting rainfall patterns occurred

during summer of 2013 and 2014 with the former showing exceptionally low and scarce rainfall, the latter high total

precipitation with rainfall events occurring frequently throughout the season.

The temporal and seasonal changes in meteorological conditions directly influenced soil conditions. Soil temperatures

responded closely to air temperature but, as would be expected, changes were buffered by the insulating effect of the soil

especially in the forest environment where it was less cold in the winter and less warm in the summer in comparison to the

air temperature (1/24°C; 4/19 °C minimum/maximum soil temperature at 5 cm depth at grassland and forest, respectively)

(Fig. 3). Weather conditions also resulted in drying and wetting of soil with the highest, relatively uniform water contents

persisting from late autumn until early spring, contrasting with very variable water contents in spring and summer. At the

forest site (T-f), especially in winter, the water content in top soil layer was distinctly higher than lower down, while at the

grassland site (T-g) the differences between SWC at different depths were less pronounced. In summer, the responses to

precipitation at different soil depths were variable: typically rainfall caused an immediate increase in SWC both in the upper

and lower soil. On some occasions (e.g. T-g 8/2013, 5/2014), however, the response of SWC to rainfall at 10 cm depth was

more pronounced than at 5 cm depth.





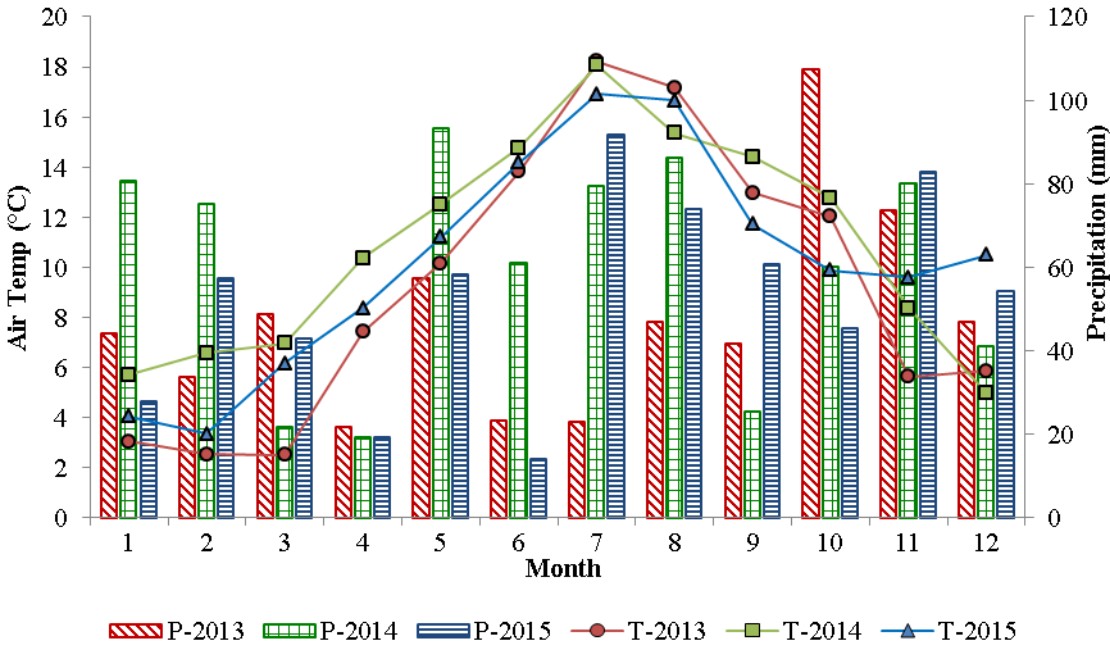

**Figure 2:** Meteorological conditions at the study sites during 3 years of measurements (2013-2015) including average monthly air temperature (T) and total monthly precipitation (P). Differently coloured bars and symbols identify each year of the measurement.





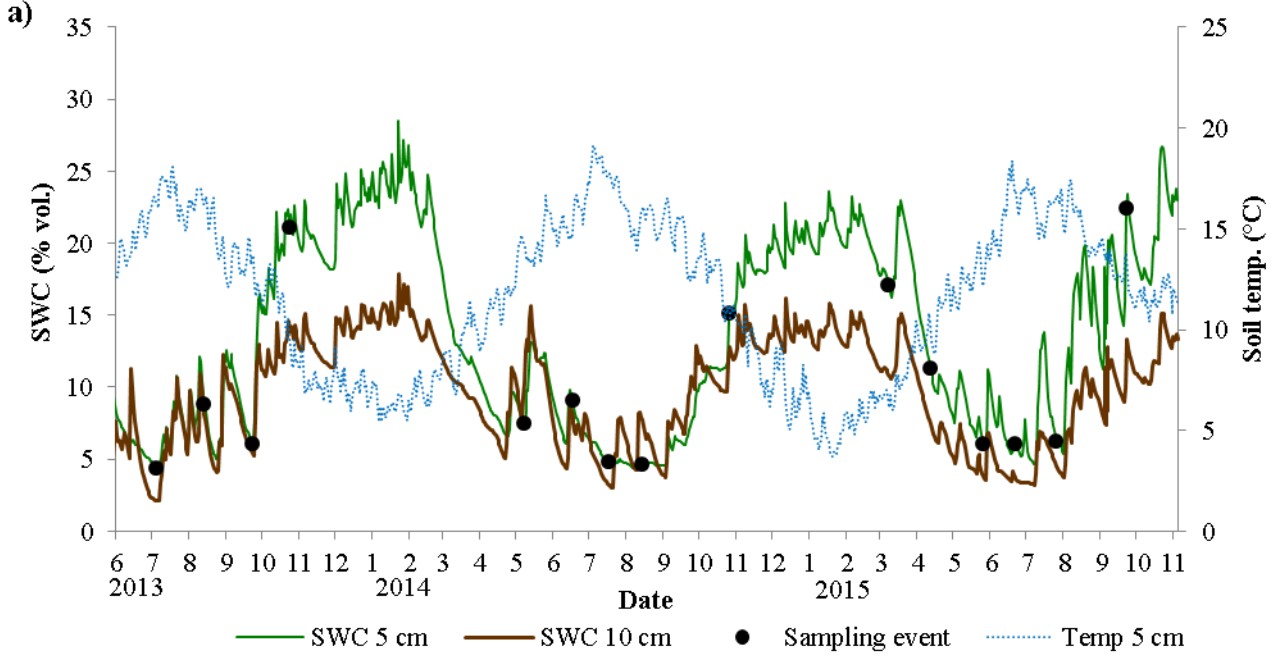

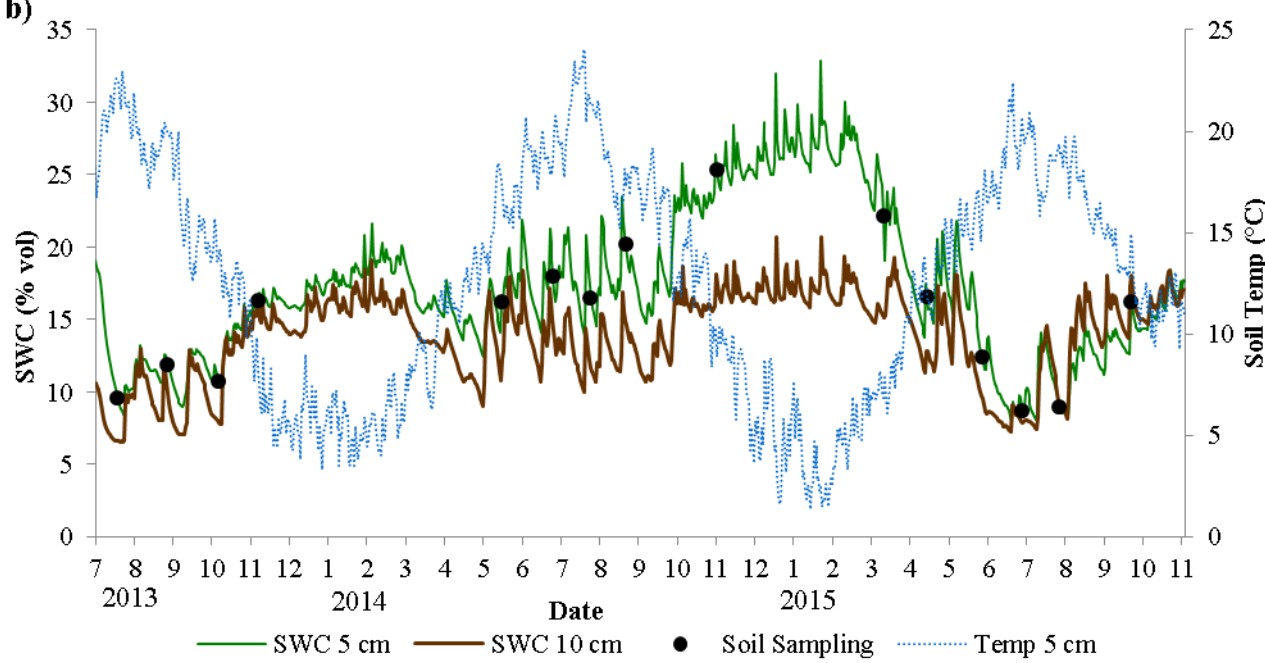

**Figure 3:** Temporal changes in soil temperature at 5 cm depth (—blue line) and soil moisture (—green line – SWC at 5 cm depth; —brown line – SWC at 10 cm depth) at both study sites over 3 years; a) Thetford-forest (T-f); and b) Thetford-grassland (T-g). Field measurements and sampling events are marked with black circles (•).

## 3.2 Seasonal changes in SWR

SWR occurred to some degree for the majority of the warmer months (May-October) followed by a change to wettable soil conditions in the colder half of the year (November - April) (Fig. 4), however, this varied from year to year depending on specific temperature and soil moisture conditions. During the warmer months of 2013 and 2015 when the total precipitation

was low, the majority of soil was water-repellent (WDPT >60 s). In 2014, during a wetter and warmer summer season, SWR was very spatially variable with parts of the soil remaining wettable (e.g. T-g 1/7/14), while the others showed moderate to slight water-repellency (WDPT 6-600 s) at site T-g and extreme to slight water-repellency (WDPT >3600 – 6 s) at site T-f. Only on a few occasions during the whole measurement period (e.g. 19/7/13 for T-g and e.g. 1/8/14 for T-f) was soil uniformly extremely water repellent (WDPT > 3600 s) which coincided with long dry spells lasting at least two weeks prior

to the measurements. For most sampling events soils showed very high spatial variability in wettability with samples exhibiting different WDPT values at each plot at a given sampling event.

The WDPT values corresponded well with SWC. Thus, for the majority of cases at lower water contents, higher WDPT values were observed, but it was also notable that highly variable SWC values were measured when soils exhibited a range of different WDPT levels.

Although the general pattern of SWR occurrence at both sites was relatively similar, soil at the forest site (T-f) showed overall higher and spatially less variable WDPT values than at the grassland (T-g) site. Thus, soil at the former site showed more frequent occurrence of extreme SWR (especially during 2014) and also a higher proportion of soil remaining water-repellent when the surrounding soil was already wettable (e.g. 9/11/13, 23/3/15, 28/4/15) (Fig. 4).





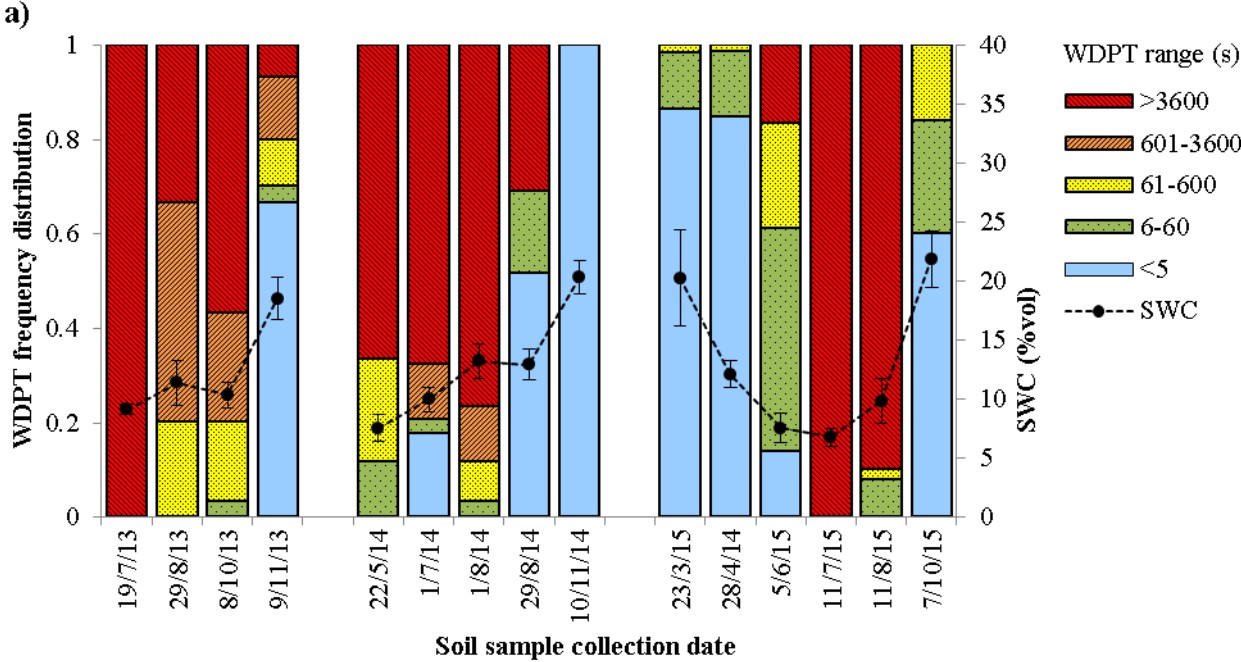

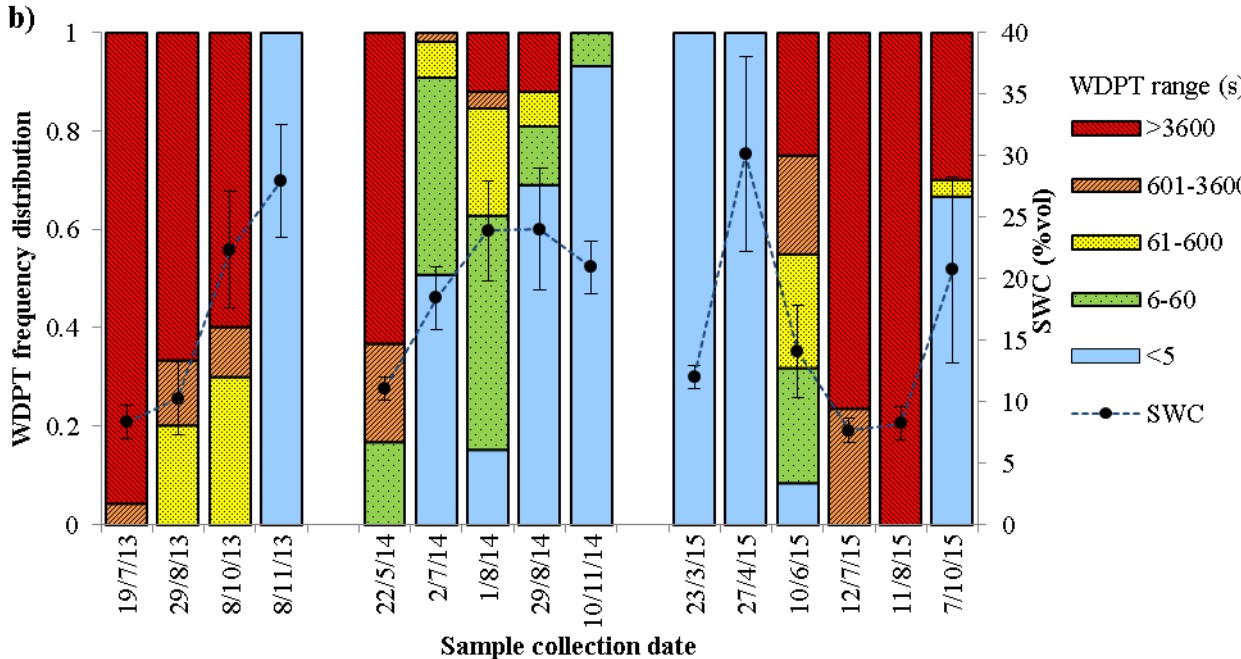

**Figure 4:** Frequency distribution of SWR persistence (measured by WDPT) and soil water content (SWC) for both study sites at 0-9 cm depth at all sampling dates (a) forest (T-f) and (b) for grassland (T-g). Different colours reflect WDPT classes, black circles represent mean SWC and error bars the standard error of the mean (n=24).



### 3.3 Seasonal variations in $CO_2$ fluxes

Measurements of $CO_2$ fluxes showed high variability between sampling events, and between the warmer and cooler periods of each year. The lowest $CO_2$ effluxes were observed in early spring (e.g. 4/15, 5/15) and late autumn (11/14), but also on a

few occasions during the summer (e.g. 7/13) (Fig. 5). The highest $CO_2$ effluxes were observed during spring and summer, which also corresponded with the highest spatial variability in effluxes between samples. Bare soil plots showed significantly lower $CO_2$ efflux than plots with vegetation and litter covers at the T-f site, but not at the T-g site (Table 2).

A clear division in soil $CO_2$ fluxes between warmer and cooler periods was observed at both study sites, highlighting soil temperature as a major factor influencing soil $CO_2$ fluxes (Fig. 6). $CO_2$ fluxes remained low up to 10 or 12 °C and increased

with rising temperature above these. Beyond a maximum around 14 °C at the forest (T-f) site and 20 °C at the grassland (T-g), however, a reduction in $CO_2$ flux was observed, with the maximum efflux being higher at the former.

The other important factor affecting soil $CO_2$ fluxes was soil moisture (Fig. 7) which, together with soil temperature, can explain overall 61% of total variations in soil $CO_2$ flux. By considering these two factors (soil temperature and soil moisture) together it was clear that especially at higher temperatures (16-20 °C), low soil moisture (SWC <20 %) can be the limiting

factor and lead to reduced soil respiration. When SWC increased, soil $CO_2$ flux was also higher, but reduced again at high SWC values. At low soil temperatures (i.e. the 8 °C temperature band), soil moisture showed a very limited effect and soil $CO_2$ fluxes remained low irrespective of SWC.

A high variability of $CO_2$ flux responses was observed even for similar mean soil water contents and the addition of other factors in the general model (e.g. study site, type of vegetation; Table 3) only slightly improved explanation of the overall

variability in $CO_2$ fluxes ($R^2$=0.68).





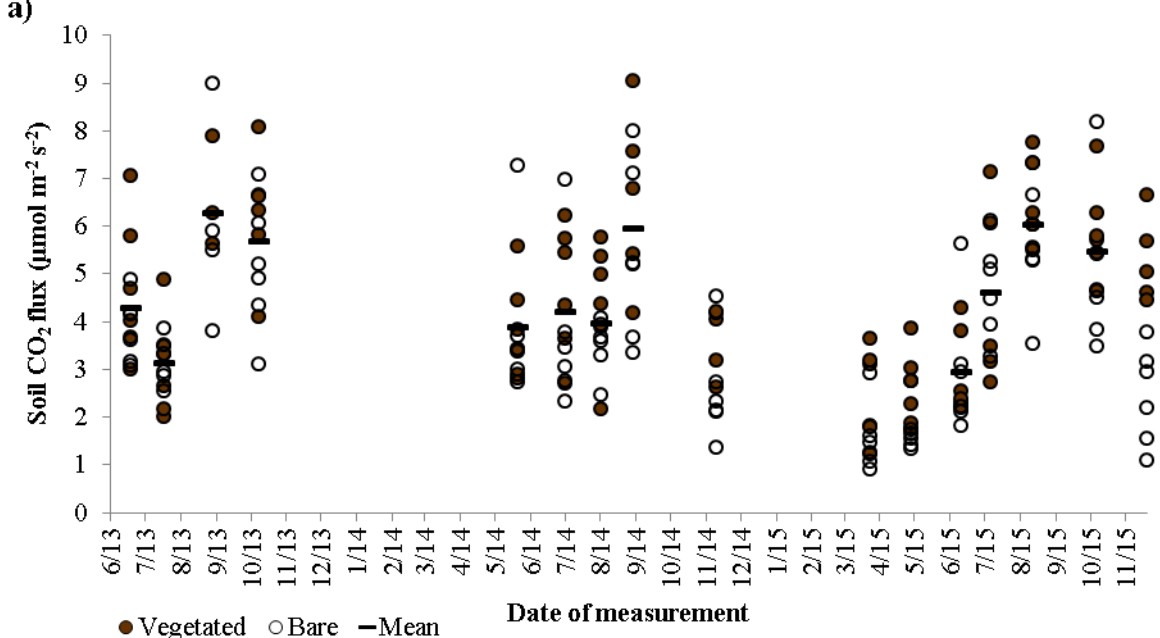

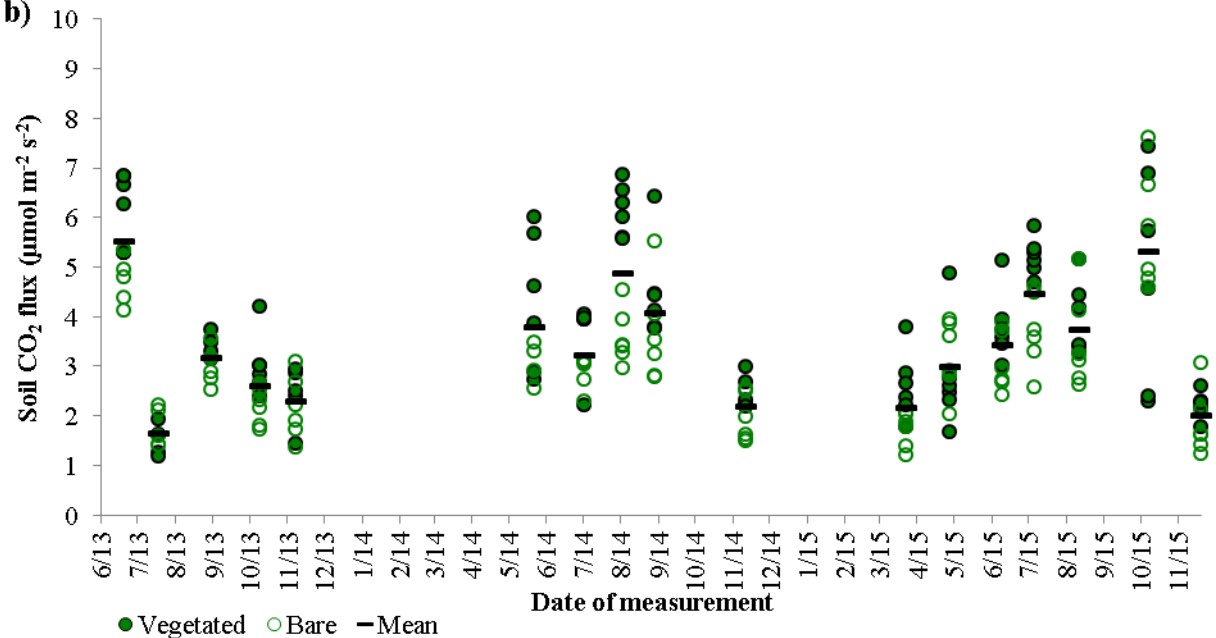

**Figure 5:** Variations in soil CO$_2$ fluxes for each measurement event for vegetated (● filled circles) and bare (○ open circles) plots at both study sites; (a) forest T-f and (b) grassland T-g.





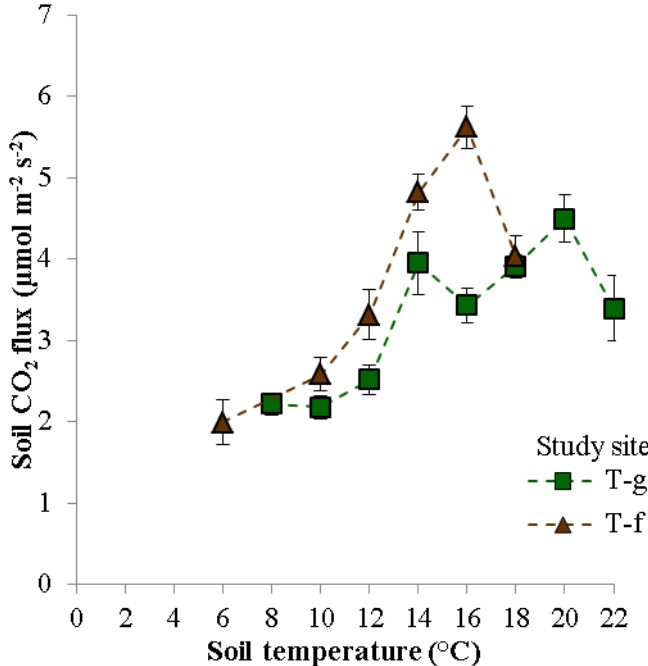

**Figure 6:** Relationship between soil $CO_2$ flux and soil temperature for the forest (T-f) and grassland (T-g) sites. Soil $CO_2$ fluxes are represented as means (with standard errors) for soil temperature grouped into 2 °C classes (±1 °C).



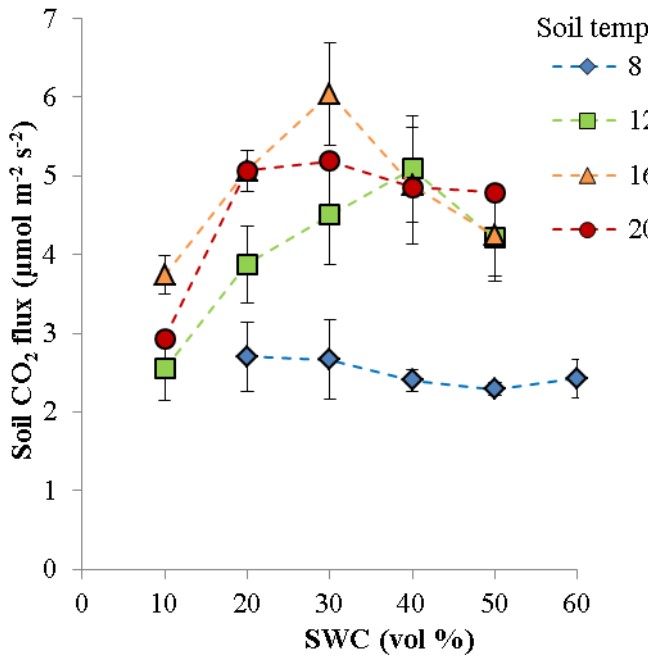

**Figure 7:** Relationship between soil $CO_2$ flux and soil water content (SWC) for the forest (T-f) and grassland (T-g) sites for different soil temperature ranges. Soil $CO_2$ fluxes are represented as means (with standard errors) for SWC's grouped into 10 % SWC. Different colours and symbols represent results grouped into 4 soil temperatures bands 8: 6.1-10 °C; 12: 10.1-14 °C, 16:14.1-18 °C and 20: 18.1-22 °C.





**Table 2:** Total average $CO_2$ fluxes ($\mu mol/m^2/s^2$) from plots under bracken and grass understorey with vegetated and bare plots at the forest (T-f) and grassland (T-g) study sites. The asterisks indicate the statistically significant differences between groups of vegetated and bare plots (*$p<0.05$, **$p<0.01$, ***$p<0.001$).

| Study site | Vegetation type | Vegetated plots mean(st.err) | Bare plots mean(st.err) |
|---|---|---|---|
| T-f | Bracken | 4.57(0.28) | 3.02(0.18) * |
|  | Grass | 5.14(0.28) | 3.93(0.27) * |
|  | all | 4.86(0.20)*** | 3.57(0.16) *** |
| T-g | Bracken | 3.61(0.23) | 3.12(0.15) |
|  | Grass | 4.04(0.22) | 2.96(0.21) |
|  | all | 3.82(0.16)*** | 3.04(0.13) *** |





**Table 3:** Factors affecting soil $CO_2$ fluxes including the statistical significance level.

| Source | Type III sum of Squares | df | mean square | F | Sig. |
|---|---|---|---|---|---|
| Corrected model for sqrt $CO_2$ flux | 23.11* | 64 | 0.36 | 3.96 | 0.000 |
| Intercept | 24.43 | 1 | 24.43 | 267.72 | 0.000 |
| SWC * Temp | 19.85 | 62 | 0.35 | 3.51 | 0.000 |
| Study Site | 1.56 | 1 | 1.56 | 17.09 | 0.000 |
| Vegetation type | 0.84 | 1 | 0.84 | 9.15 | 0.003 |
| Error | 10.86 | 119 | 0.09 | | |
| Total | 788.60 | 184 | | | |
| Corrected total | 33.96 | 183 | | | |

\* $R^2 = 0.68$




### 3.4 Soil water repellency and CO$_2$ fluxes

Given that soil wettability was strongly affected by both temperature and moisture, SWR effect on CO$_2$ fluxes was therefore considered separately from the above described model (Table 3). A more hydrologically meaningful analysis of the potential role of SWR was carried out by separating the results into groups representing the relative fraction of extremely water

repellent soil (WDPT >3600 s) for each sampling event (Fig. 8). This grouping of SWR results was used as a proxy of heterogeneity of soil moisture distribution in soils affected by water repellency. The zero value represented completely wettable soil where water distribution was not affected by water repellency, and a value of 1 denoted uniform extremely water repellent soil where similarly low moisture content was expected throughout the soil. Values between zero and 1 represented increasing levels of extreme SWR presence; lower values indicated wettable soil with isolated patches of

extremely water repellent soil, while the values closer to 1 represented soils dominated by extreme water repellency with isolated zones of wettable soil or low SWR.

Soil CO$_2$ flux showed a very clear response to SWR distribution. When SWR distribution had a value of zero (i.e. the entire soil was wettable) soil CO$_2$ flux was low, but it increased when a small fraction of soil became extremely water repellent. The maximum soil CO$_2$ flux was reached for a SWR distribution around 0.4 – 0.6. SWR distribution values >0.6 were

associated with a decreased CO$_2$ flux, which reached its lowest values when all soil became uniformly water repellent (value of 1). The differences between soil CO$_2$ fluxes for wettable/extremely SWR distribution values (0 and 1) and intermediate values (0.2-0.8) were observed mainly for events with higher soil temperatures and in many cases they were statistically significant. Considering the whole soil volume examined here, we can therefore reject the hypothesis that presence of water repellency unequivocally reduces soil respiration also under 'real world' field conditions. The response of soil respiration to

the presence of soil water repellency is more complex than it has been originally anticipated and its effects are clearly more complex as discussed below.





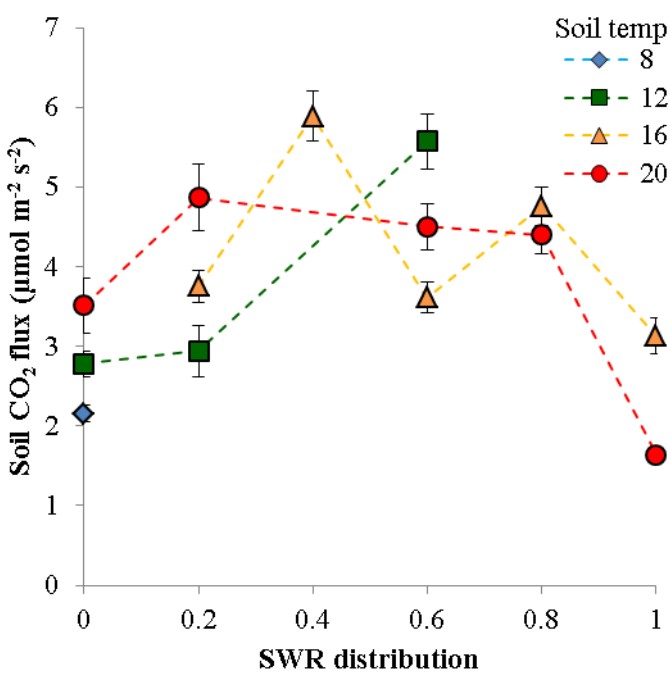

**Figure 8:** Soil $CO_2$ flux response to SWR distribution (0=wettable, 1=uniformly extreme SWR) for different soil temperature bands (8 – 6.1-10 °C; 12 – 10.1-14 °C, 16 -14.1-18 °C, 20 – 18.1-22 °C). Soil $CO_2$ fluxes are represented as means (with standard errors) for SWR distribution grouped within ±0.1.





## 4 Discussion

### 4.1 Temporal variations in SWR

This study investigated, for the first time, the seasonal variability of water repellency persistence in UK soils and, for the first time globally, the potential impact of associated soil moisture distribution on $CO_2$ fluxes in the field. Three years of

monitoring of soils under humid temperate pine forest and grassland in England, revealed that SWR was present for most of the spring, summer and autumn. The presence of SWR at these locations was consistent with previous studies that also reported severe SWR for UK grassland, forest and heath (Doerr et al., 2006), arable land (Robinson, 1999; Hallett et al., 2001) and on golf greens (York and Canaway, 2000), and in The Netherlands on grass-covered sand dunes under a similar climate (Dekker and Ritsema, 1996a; Ritsema and Dekker, 2000). Both investigated sites were under permanent vegetation,

which is generally considered to be situation most susceptible to SWR development (Doerr et al., 2000; Woche et al., 2005) due to the continuous input of hydrophobic substances from the vegetation and soil microbes (Doerr et al., 2000), and a low level of soil disturbance.

SWR has long been known to be temporally variable and has commonly been observed during warm dry conditions, while disappearing during prolonged cold and wet conditions (Doerr et al., 2000; Leighton-Boyce et al., 2005; Buczko et al., 2006;

Stoof et al., 2011). At the sites investigated here, SWR was observed from early summer (May/June) until late autumn (November). The exact timing of water repellency development and also of its complete disappearance could not be precisely pinpointed in this study within each year due to the monthly timings of the sampling visits, but it was clearly associated with low soil moisture contents and higher soil temperatures. SWR was not observed at soil temperatures lower than 10 °C despite low soil water contents, suggesting not only soil moisture but also the temperature is important in SWR

development. SWR remained spatially and temporally variable throughout the entire warmer periods. Only long dry spells resulted in high persistence of water repellency being uniformly distributed in the entire soil. For the majority of the warmer season, SWR was present, but of variable severity and often spatially interspersed with a small proportion of wettable zones. The high variability of SWR can be attributed to frequent change between and sufficiently dry and wet periods, characteristic of the UK climate, which allow development and partial disappearance of SWR. During the warmer dry periods in 2013 and

2015, the data suggest that soil became water repellent throughout (WDPT >5 s), but its persistence in different soil areas



varied from minutes to hours. In contrast, during summer 2014, the proportion of wettable soil patches near water-repellent zones was very high (up to 65 % in T-g, up to 50 % in T-f), which can be attributed to the particularly rainy summer (total rainfall for summer 2014 was 50 % higher than 2013, 20 % higher than 2015). The high spatial variability of water repellency and that partial change to a wettable state during the summer is likely to be a consequence of spatially uneven infiltration into the soil and further enhanced by preferential flow, both caused by presence of hydrophobic particles surfaces. The flat topography and surface cover of litter (at the forest site) or vegetation (at the grassland site) probably restricted surface runoff and resulted mainly in spatially variable infiltration and preferential water flow (Bughici and Wallach, 2016). Most rainfall was likely transferred below the near-surface repellent layer via preferential flow zones formed by faunal burrows (Shakesby et al., 2007), roots and soil cracks (Dekker and Ritsema, 1996b; Kobayashi and Shimizu, 2007; Urbanek et al., 2015). The preferential flow paths induced by SWR have most likely resulted in the high spatial variability of water repellency and water content of the soil, as it is known that the soil adjacent to preferential flow paths is the first zone of the soil to switch into a wettable state (Urbanek et al., 2015). SWR induced preferential flow caused creation of dry, isolated water repellent soil patches that were frequently detected on occasions when the majority of soil was wettable (Fig. 4). These isolated dry soil patches would have been not only deficient in water, but would also have had a restricted supply of nutrients, due to the lack of their transfer by water (de Jonge et al., 2009; Goebel et al., 2011).

## 4.2 Temporal variations in soil C fluxes

Temporal fluctuations in soil temperature and moisture not only affected the presence or absence of SWR, but were likely to be also responsible for the variability in soil respiration and C fluxes. The $CO_2$ flux measurements at the study sites were conducted each year from June until November with only a few early measurements in spring during 2015. Thus no information is available on soil respiration during the winter season. All early spring and late autumn measurements, however, showed lower soil respiration rates than during the warmer period. During the colder and typically wetter part of the year, primary productivity, soil biological activity and therefore soil respiration is typically low (Davidson and Janssens, 2006). Considering the seasonal fluctuation, but also noting the positive correlation between soil $CO_2$ fluxes and soil temperatures, it is clear that the latter constitute the main factor affecting soil respiration, which is consistent with many



previous studies (Gaumont-Guay et al., 2009; Yvon-Durocher et al., 2012; Karhu et al., 2014). The positive response of $CO_2$ flux to increasing soil temperature reflects the greater activity of roots and decomposing microorganisms, but can also involve long-term changes in microbial population communities and higher substrate supply from photosynthesis in response to longer-term trends as expected, for example, with global warming (Davidson and Janssens, 2006; Gaumont-Guay et al., 2009). At both study sites soil respiration increased with rising temperatures, but only until a maximum level was reached, after which a notable decrease was observed. The occasions when soil $CO_2$ fluxes were no longer dictated by temperature occurred during the summer when the soil was exposed not only to relatively warm, but also dry conditions for prolonged periods, suggesting that soil moisture was the restricting factor. The effect was observed only at times of uniformly low soil water contents when persistence of SWR was consistently high. On the occasions of measurements with low, but spatially variable water content, soil respiration was high and followed an increasing trend with temperature. A reduction in soil moisture availability is known to reduce microbial activity and root respiration (Or *et al.*, 2007). Prolonged summer droughts have been recognised in many studies as the cause of a decrease primarily in heterotrophic respiration which, according to Borken et al. (2006) could cause increases in the storage of soil organic C in this forest type.

## 4.3 Effect of soil moisture and SWR on soil $CO_2$ fluxes

Soil $CO_2$ fluxes were found to respond to changing soil moisture content particularly at higher soil temperatures (Fig. 7), but the variability in $CO_2$ flux remained high especially for intermediate soil moisture contents. Only after long dry spells when soil moisture availability was low, were soil respiration rates significantly reduced. At high soil water contents, soil $CO_2$ efflux was high but also very variable most likely due to frequent wetting and drying events resulting in very heterogeneous soil moisture distribution (Gaumont-Guay et al., 2009). The variable soil water distribution and inconsistent response in soil respiration with temperature and moisture content, can be explained by the presence of SWR, which is known to substantially affect soil water distribution and thus processes where water is involved (de Jonge et al., 2009), including microbial activity and therefore soil respiration. Some previous studies have already shown that SWR can protect C from decomposing microorganisms (Goebel et al., 2005; Goebel et al., 2007; Bachmann et al., 2008; Lamparter et al., 2009; Goebel et al., 2011) and result in reduced soil respiration. These laboratory-based studies focused mainly on the severity of

SWR of homogeneous soil and therefore did not explore the wide range of scenarios to which natural soil is exposed. Most studies exploring SWR present the results based on overall median or mean WDPT values, which does not allow identification of the naturally rather common and hence important condition when SWR variability is very high. Presenting water repellency distribution rather than the mean or median value (Fig. 8) is therefore hydrologically more meaningful. It includes soil wettability conditions with uniformly low (wettable) and high (extreme) water repellency, but also identifies the intermediate stages when soil is dominated either by wettable soil with patches of extremely water repellent soil or *vice versa*.

The results demonstrate for the first time that (i) there are different responses of soil $CO_2$ fluxes to different patterns of SWR distribution (i.e. SWR does not simply reduce soil $CO_2$ fluxes) and (ii) that the effects are consistent across a range of temperatures. Based on these findings, we present a new conceptual model for $CO_2$ flux behaviour (Fig. 9) that accounts for the more realistic effect of SWR observed in this field study and includes three main SWR-sensitive hydrological condition.

Wettable soil (Fig. 9a), represents a condition observed when a soil water repellency is absent due to frequent wetting events and therefore high soil moisture contents, or a situation when the temperatures are too low for water repellency to develop. Under these conditions, soil water is relatively uniformly distributed and soil pores are either fully or partly filled with water. Owing to low temperatures and/or high soil water contents microbial activity is limited resulting in low soil $CO_2$ production. Water-filled pores also result in restricted gas exchange between the soil and atmosphere and thus low $CO_2$ efflux.

Uniformly extreme water repellent soil (SWR distribution equal to 1) (Fig. 9c) is associated with consistently low moisture content and soil $CO_2$ fluxes. Several laboratory studies have reported low respiration rates in similarly highly water repellent soil (Goebel et al., 2007; Lamparter et al., 2009). Owing to low water availability, microbial and enzymatic activity is reduced (Or et al., 2007; Moyano et al., 2013; Moyano et al., 2012), or it ceases entirely when extremely low matric potentials are reached and water films in soil pores become disconnected (Goebel et al., 2007). According to Or et al. (2007), diffusion rates of extracellular enzymes produced by microbes to access organic matter are proportional to the thickness of the water film surrounding soil particles and this thickness is substantially reduced by SWR (Churaev, 2000; Goebel et al., 2011). Obstruction of microbial movement and reduction in diffusion results in physical separation of microorganisms from substrates and nutrients, which can lead to long-term starvation (Kieft et al., 1993). At the sites investigated, such a situation



was observed only on a few occasions following long dry spells, suggesting that under the current humid-temperate, this soil condition is rare here. Uniformly high SWR is, however, very common in climates with distinct dry seasons or more prolonged dry periods (Doerr et al., 2003; Doerr and Moody, 2004; Leighton-Boyce et al., 2005; Stoof et al., 2011) and may become more common in the future in the UK according to climate predictions (IPCC, 2013).

The third, intermediate situation, which is examined in this context for the first time here, is the hydrological status of variably water-repellent soil (SWR distribution 0.2-0.8) where soil is dominated by wettable or water-repellent soil patches. In a humid temperate climate with soils susceptible to SWR, this likely to be the most common soil condition, while in climates with distinct dry seasons or common dry spells it represents the state of change between wettable and water-repellent taking place between the wet and dry seasons or periods (Leighton-Boyce et al., 2005; Stoof et al., 2011). Under

such conditions, soil is exposed to pronounced preferential flow where water infiltrates the soil via selected zones, leaving other areas completely dry (Fig. 9b). Supply of water and nutrients in these flow paths is very high and soil areas near flow paths harbour larger bacterial densities (Vinther et al., 1999) and activities (Pivetz and Steenhuis, 1995) than the adjacent soil matrix. The strongly water-repellent soil zones near flow paths with air-filled pores provide routes for gas transfer where the $O_2$ and $CO_2$ released by the decomposing microorganisms can easily be exchanged between the soil and atmosphere (Or

et al., 2007; Kravchenko et al., 2015). Very favourable conditions for microbial respiration, as well as gas exchange through air-filled pores parallel to preferential water paths, thus allow the highest $CO_2$ efflux. Understanding of soil respiration under the intermediate status of SWR distribution shows that SWR reduces soil respiration only under very extreme uniform SWR conditions whereas, when enhanced preferential flow is encouraged by hydrophobic particle surfaces, the opposite effect applies.

Future studies investigating C dynamics in water repellent soil are still needed to explore further the effect of hydrophobic particle or soil pore surfaces on soil $CO_2$ fluxes. For example, further insights could be gained by more frequent or near continuous monitoring of soil respiration together with SWR and soil moisture. This would allow better understanding of soil respiration during the wetting and drying processes in soils that exhibit SWR and thus restricted water infiltration. We consider the proposed conceptual model depicted in Fig. 9 to be sufficiently simple to be fundamentally applicable to a wide

range of water repellent soils. However, given the potential importance of SWR to affect soil respiration and ultimately soil



C storage under changing land uses and a changing climate, further field investigations involving different soil types and

environmental conditions would be valuable in determining how widespread and temporally common this scenario is.





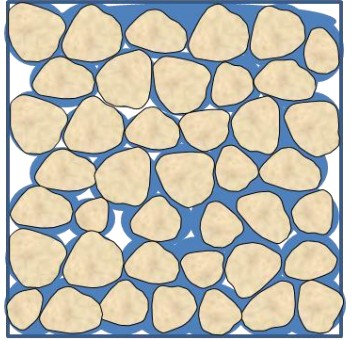 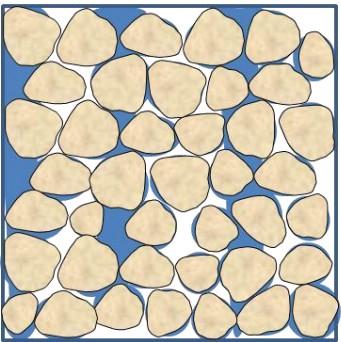 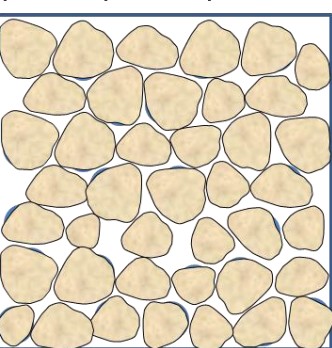

**a) Wettable soil**

- Water distribution – uniform, unaffected by water-repellency
- Pores fully/partly filled with water
- Gas exchange – low
- Microbial respiration - low

**b) Variably water-repellent soil**

- Water distribution – variable; enhanced preferential water flow
- Good gas exchange, especially in air- filled tunnels
- High microbial respiration in areas with easy water/nutrient access

**c) Uniformly water-repellent soil**

- Limited water access
- Soil pores filled with air
- Very good gas exchange
- Low microbial respiration

**Figure 9:** Soil $CO_2$ flux responses under three distinct hydrological situations associated with different soil water repellency (SWR) states and their associated soil water distribution.



## 5 Conclusions

This study reports for the first time how seasonal changes in SWR distribution affect soil respiration and demonstrates that the presence of SWR does not simply lead to a reduction in soil $CO_2$ efflux. The sites investigated in the UK under grassland and pine forest exhibit a strong presence of SWR during warmer periods, which is also dominated by high spatial variability

in SWR persistence. Frequent wetting and drying events, common in humid-temperate climates, result in high patchiness of SWR, and only when soil is exposed to longer dry spells does it become severely and uniformly water repellent. As the hydrological consequences of variable SWR spatial distribution are unique, it is necessary to recognise their distinctiveness as well as the hydrological conditions associated with entirely wettable or water-repellent soil. The data collected here suggest that the response of soil $CO_2$ efflux strongly depends on soil wettability status and the distribution of water-repellent

patches. Very high SWR levels throughout are indeed is associated with low soil $CO_2$ efflux, caused by reduced $CO_2$ production by water-stressed microbial communities. However, variable SWR distribution, results in the highest $CO_2$ fluxes, most likely due to microbial communities being concentrated in the water and nutrient 'hotspots' bordering preferential flow paths coupled with and very favourable gas exchange conditions in hydrophobicity-controlled air-filled pores. A wettable soil state only occurred at the study sites when soil temperatures were low or there was high frequency of rainfall events, and

was associated with low $CO_2$ fluxes. The hypothesis that presence of water repellency unequivocally reduces soil respiration, also under the 'real world' field conditions examined for the first time here, is therefore rejected.

SWR clearly has an important effect on soil respiration, but its impact is more complex than previously assumed, with its spatial variability likely to be the most influential factor. The presence of SWR can not only reduce soil respiration in affected soil zones. It can actually lead to enhanced respiration from soil zones exhibiting high spatial variability in SWR.

When examining SWR, this should therefore not be restricted to simply recording whether soil is wettable or water repellent with a certain persistence or severity level. Its spatial (and temporal) variability is of paramount importance. This combined knowledge should then allow prediction of the response of soil respiration to different temperature and moisture conditions.

In view of current climatic predictions and expectations that SWR will become even more widespread globally than is the case at present, it is important to include analysis of the spatio-temporal characteristics of SWR in long-term respiration





studies so that a comprehensive understanding of the specific effects of SWR on soil C dynamics under current conditions can be gained and a firmer foundation for prediction under future climatic scenarios can be established.

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
