# Peer review of "Spatially variable soil water repellency enhances soil respiration rates (CO2 efflux)"

_Biogeosciences, 2017_

## Referee Comment (RC1) · Anonymous Referee #1 · 5 Apr 2017

General comments:

This study focused on the impact of water repellency (SWR) dynamics on soil respiration. During the growing season of three consecutive years the authors monitored water content, temperature, SWR and CO2 fluxes in topsoils of a grassland and a pine forest site located in eastern England. SWR was quantified in field-moist state with the commonly used water drop penetration time (WDPT) test. In order to parameterize the heterogeneity in SWR distribution the authors derived a parameter representing the relative fraction of extremely water repellent soil (i.e. WDPT > 3600 s). The results revealed a variable distribution of SWR with large temporal changes during the growing season which was explained by variation in soil moisture and temperature. The main

outcome of the study is that the derived SWR distribution parameter was found to be associated with the measured CO2 fluxes, where the highest respiration rates were measured for variably water repellent soil.

There is a couple of studies emphasizing the potential role of SWR in explaining lag-effects of soil respiration observed after experimental droughts, but so far there is no study in which these effects were directly investigated in the field. In this regard the study by Urbanek and Doerr is timely and would be of potential interest to the readers of Biogeosciences.

However, in its current state, there are several issues with this generally well-written manuscript that need to be addressed. The most important concern I have with the manuscript is that due to the strong co-correlation between soil temperature, soil water content and SWR it is not clearly distinguishable whether the observed effects on CO2 efflux were due to temperature/soil moisture or SWR. This is, of course, a general problem with field studies and the reason why investigations on this issue are usually performed in the lab where the temperature and soil moisture effects can be controlled for.

Another crucial problem with this study is that SWR was determined only for the topsoil while soil respiration arises from the whole soil, making it hardly possible to directly relate these parameters. It is questionable what effect a heterogeneous distribution of SWR in the topsoil (0-10 cm depth) would have in the subsoil. So speculation about possible formation of preferential flow paths is not warranted, particularly when no information on subsoil SWR is available. Even if topsoil SWR is very heterogeneous it might have only a low impact on subsoil water distribution when the subsoil is completely wettable. It would therefore be necessary to investigate both SWR and soil moisture dynamics also in the deeper soil to be really able to infer their impact on soil respiration, consisting of both microbial and root respiration. Although SWR certainly affects the soil moisture pattern, it is soil moisture that actually controls soil respiration. There are several assumptions that are not justified based on the experimental findings

of the study as well as inconsistencies in the discussion. Therefore, I would not support publication of the manuscript in its current form but encourage its resubmission after a substantial revision according to the points given below. It would certainly help to improve the manuscript if the results are treated and presented as being the outcome of a case study, meaning that a generalization of the observed effects is not necessarily possible.

Specific comments:

Title: The title states that spatially variable water repellency enhances soil respiration. This is not correct because it is not SWR itself but rather the (SWR-affected) soil water content (and temperature) that actually controls soil respiration. Replacing 'enhances' by 'is associated with high' would therefore be more appropriate. Moreover, using the term 'spatially' in the title is somewhat misleading as it suggests that the study was focused on the spatial distribution of SWR at the study sites. However, deriving conclusions about the spatial distribution of SWR is simply not possible based on the investigation of only six soil cores per site.

P1L7: Here, hydrophobicity is used as a synonym of soil water repellency. This is not correct because SWR covers the entire range of states where soil repels water, while hydrophobicity explicitly denotes a state where water is not able to penetrate the soil (often defined as having a soil-water contact angle above 90 degrees).

P1L18: The authors discuss preferential flow as a possible mechanism to explain their results. This is fine in the main text, however, as this was not proved in the study it is conjecture and should not be in the abstract.

P4L6: What is meant by 20-m transect here? Is 20 m the distance between the plots on the left and the plots on the right? If yes, then including a scale would certainly help the reader because it is not immediately intelligible from Fig. 1 that the plots are arranged along a transect.

P7L18-20: Given the total number of measurement events (n = 16) I was wondering whether the removal of soil material approx. 10 cm away from the flux collars would not influence the moisture distribution and hence CO2 efflux. Could you please comment on that.

P8L7-8: The determination of WDPT frequency distribution and the SWR distribution parameter was based on measurements carried out on material from 4 depths at 6 plots. While SWR distribution with depth could be reasonably described, this is clearly not possible for the horizontal distribution as the plots were located several meters away from each other, not allowing to draw meaningful conclusions regarding the spatial dependence and spatial structure of SWR. Moreover, considering that the material for the SWR determination was extracted at some distance from the flux collars, it seems very difficult to directly relate the measured CO2 fluxes to the measured SWR distribution.

P12L18: What is meant by 'surrounding'? As the plots are several meters away from each other, it is not possible to draw any conclusion about the conditions of the surrounding soil (i.e. in close proximity).

P13, Figure 4: What is the rationale for using the standard error here (and in Figures 6, 7, 8 and Table 2)? Using the standard deviation (as in Table 1) is more appropriate to get an idea about the variation of the water content.

P20L5: The authors assume that the SWR distribution parameter can be used as a proxy of heterogeneity in soil moisture distribution in the flux collars, however, the validity of this assumption was not proved in this study and seems highly questionable considering the points mentioned above.

P20L8: The assumption that uniformly water repellent soil (SWR distribution = 1) is necessarily associated with homogeneously distributed low moisture content is not valid. This becomes immediately evident when considering that the calculation of this parameter is based on core material extracted from plots that were located several me-

ters away from each other. Considering the dimension of the soil cores (5 cm diameter, 9 cm length) it becomes clear that the SWR distribution parameter is not representative of the site and not even representative of the individual plot. In other words, it is easily conceivable that the wetting properties and thus the moisture distribution of the surrounding soil is different from that measured for the soil cores.

P22L21-22: Such detailed statements regarding SWR distribution at the sites are not justified (see comments above).

P23L3-5: Apart from the fact that spatial heterogeneity was actually not investigated in the present study (this is simply not possible by investigating only six soil cores per site) this statement is difficult to understand and in contrast to the assumption that SWR is the cause of preferential flow and a heterogeneous water distribution as stated, for instance, at P26L9-11. What is the authors' opinion? Is spatial variability of SWR caused by a spatially uneven infiltration into the soil which, in turn, is affected by preferential flow, or is SWR itself the cause of an uneven water infiltration and preferential flow phenomena?

P23L10-12: The statement in this sentence is not clear (see comment above). It is not proper to state that the preferential flow paths caused by SWR resulted in a high spatial variability of SWR.

P24L18-20: The statement in this sentence (high $CO_2$ flux at high water content) is in contrast to the findings presented in Figure 7 and the conclusions and are not consistent with the 'model' presented.

P24L25: What is meant by 'severity of SWR'? Is it different from 'persistence of SWR'?

P25L8-10: The use of 'response' is not justified in this context because it is not SWR itself but rather the SWC (influenced by SWR) that actually influences soil respiration. Using 'associated' would be more appropriate ('... different $CO_2$ fluxes were associated with different patterns of SWR ...').

P25L11: Please check this sentence. What is meant by '... the more realistic effect of SWR ...'? (more realistic than ... ?).

P25L12: I have some issues with the 'conceptual model' presented in Figure 9. According to the model, wettable soil (Figure 9a) represents a condition where soil moisture is too high or soil temperature is too low for SWR to develop. The $CO_2$ efflux associated with this particular state was found to be low. However, it was not SWR that caused the low $CO_2$ efflux but rather the high water contents or the low temperatures (as was correctly stated by the authors). Hence, it is not justified to state that the model is accounting for the complex effect of SWR as both SWR and $CO_2$ efflux are simply co-correlated and controlled by soil moisture and soil temperature. In addition, Figure 9c, which represents the 'water repellent state' with uniformly water repellent soil suggests extremely low water contents (near zero) as compared to the other states. Apart from the general problem of relating the measured parameters in the present study (please see comment to P20L8), the results presented in Figure 4 show that this is not necessarily the case. As shown in Figure 4a, there was a transition from a uniformly water repellent soil (on 19/7/13) to a variably water repellent soil (on 29/8/13 and 8/10/13), while the corresponding water content remained fairly constant around 10 vol-%, which is far from being completely dry (as suggested in Figure 9c). There is also some ambiguity about the intermediate (variably water repellent) state illustrated in Fig. 9b. What do the authors really think? Is SWR the cause of an uneven water infiltration and causes preferential flow phenomena, or is it the spatially uneven infiltration into the soil which, in turn, is affected by preferential flow that causes the high spatial variability of SWR (as stated at P23L3-5)? Generally, the proposed 'model' would only be valid for the specific conditions of the sites investigated. For instance, it is well conceivable that a wettable soil is characterized by an intermediate water content (particularly in case of sandy soils). And the occurrence of such a situation is also possible in summer as shown in a study by Buczko et al. (2007, Ecological Engineering 31: 154–164). Under such conditions (i.e. intermediate water contents and high temperature) microbial activity and $CO_2$ efflux can be expected to be high (and might be even higher than for

variably water repellent soil). Overall, given the lack in general validity and explanatory power, using the term 'model' seems not appropriate, although the given explanations and the illustrations in Fig. 9 are valuable for understanding the observed effects on CO2 efflux at the investigated sites.

P26L16-19: Again, it is not reasonable to state that the intermediate state of SWR enhances soil respiration. It is indeed conceivable that CO2 efflux of a wettable soil, which is characterized by an intermediate and homogeneously distributed water content, is even higher than of a variable water repellent soil, provided that the temperature is high enough (see comment above and comments to P25L8-10 and the title)

P29L10-19: The conclusions presented here are not justified (see comments above).

Other minor points:

P1L12: SWR is introduced at P1L7 and should subsequently be used instead of 'soil water repellency' throughout the text. This should be checked carefully as there are many instances where 'soil water repellency' or 'water repellency' is used.

P2L5-7: The statement that soil moisture controls pore-water connectivity is self-evident and should be removed.

P3L4: SOC is introduced at P2L6 and should subsequently be used instead of 'soil organic C' throughout the text.

P3L18: Please check the style of the sentence (..., which ....., which).

P4L8: Please replace 'for' by 'at' (At each study site ..., and at each ...)

P6, Table 1: Please replace 'for' by 'of' (Selected soil properties of samples ...)

P7L6: Please replace 'for' by 'at' (At each study plot ...) and 'was' by 'were' (... soil collar were temporarily removed ...).

P7L12: I would suggest to replace the sentence by: '... was determined by fitting an

exponential function to the evolution of CO2 concentration over time ...'.

P7L13-14: Please check the style of this sentence. In addition, could you please add information about the overall percentage of fittings with $R^2$ <0.95.

P7L19: Can you please specify what is meant by 'further soil measurements'.

P8L3: Was bulk density really determined after each field visit?

P8L7-8: Could you please state how many replicate measurements per plot and depth were carried out.

P8L18: Please check this sentence ('... to reduce oxides of N, CO2 and N2 were determined...')

P8L20: 'distilled' or rather 'deionized' water?

P8L22-23: Please use SWC instead of 'soil water content'. This should be checked carefully throughout the text.

P9L2-3: Could you please state the post-hoc test used in conjunction with the ANOVA.

P10, Figure 2: Please replace 'Air Temp' by 'Air temperature'.

P11, Figure 3: Consistent labeling should be used ('Soil temperature', 'vol-%'). Please use either 'Sampling event' or 'Soil sampling' in the legend. Is it correct that Fig. 3a begins with June 2013 while Fig. 3b begins with July 2013?

P12L6-7: Please use the same rank order for text and numbers (from low to high), i.e., '... slight to moderate (WDPT 6 to 600 s) ...' and '... slight to extreme SWR (WDPT 6 to >3600 s).

P13, Figure 4: Please be consistent with the labeling used in Figure 3 (vol-%) and use the same labeling for a and b (either 'Soil sample collection date' or 'Sample collection date'). Please use site designations consistently throughout the text and figures. Currently there are several variants, e.g., forest (T-f), forest site (T-f), Thetford-forest (T-f),

Interactive
comment

etc.

P14L10: Is 14°C correct? Figure 6 shows the highest fluxes at the forest site to be around 16°C. Is there any explanation for the large difference in temperature where the maximum CO2 fluxes were found?

P17, Figure 7: Please insert '(°C)' after 'Soil temp.'. 'temperature ranges' -> 'temperature bands'. Please replace '... for SWC's grouped into 10% SWC ...' by '... for SWC grouped into classes of 10 vol-% ...'.

P18, Table 2: The case '**p<0.01' does not appear in the table and should be removed.

P19, Table 3: Using * for referring to the footnote is not appropriate here as * is also used in the interaction term 'SWC * Temp'.

P19L18-21: This paragraph is not adequate in the Results section and should be moved to the Discussion.

P21, Figure 8: Please insert '(°C)' after 'Soil temp.'. Please use a consistent description of the temperature bands in Figure 8 and Figure 7 (P17).

P22L6 and L15: These statements here are inconsistent ('SWR was present for most of spring, summer and autumn' vs. 'SWR was observed from early summer until late autumn').

P22L23: Please delete 'and' in this sentence to read: '... frequent change between sufficiently dry and wet periods, ...'.

P22L24: Please change to '... which allows development ...'.

P23L3: Please replace the comma by 'and' to read: '... higher than 2013 and 20% higher than 2015.'.

P23L19: What is meant by C fluxes here? Referring to soil respiration would be sufficient here as no other C fluxes (e.g. transport of dissolved organic matter) were

investigated in the present study.

P24L13: The reference is lacking: what is meant by 'this forest type'? This needs to be specified.

P24L16: Using 'but' in the context of this sentence is not appropriate.

P25L5-7: Please check the style of this sentence ('... wettability conditions with uniformly low (wettable) and high (extreme) water repellency ...' as well as '... when soil is dominated either by wettable soil ...').

P25L12: Please check this sentence ('Wettable soil ... represents a condition observed when a soil water repellency is absent . . .').

P29L6: Please add an 's' to read '... becomes severely ...'.

P29L10: Please change to '... were indeed associated ...'.

---

## Referee Comment (RC2) · Anonymous Referee #2 · 5 May 2017

The manuscript 'Spatially available soil water repellency enhances soil respiration rates (CO2 efflux)' by Urbanek and Doerr describes a 3 year study investigating the effect of soil moisture and soil water repellency on soil respiration. Two study sites with similar soil type were chosen in England. Plots of vegetated and bare soil were measured for soil moisture, soil temperature and soil water repellency. The authors observed temporal variability in soil water repellency during the year. They aim to link the observed soil water repellency to soil water and temperature, as well as to soil respiration. Lastly, the authors present a theoretical framework to explain the observed link between soil moisture and soil respiration.

The study is interesting and addresses an interesting phenomenon: the effect of soil

moisture repellency on soil respiration. The authors present a lot of data, which I find overwhelming. In my opinion, the manuscript would benefit from focusing, especially of the results section. Data that are not crucial for the explanation could go into supplementary information. This would guaranty that it is not an information overflow. General questions that need clarification:

- The experimental setup as described in Figure 1 and Table 1: I have a hard time following why bracken and vegetated soil was measured, and also what the information on bare soil and vegetation soil was used for. E.g. Figure 4 presents data from which plots exactly? And Figure 5 displays forest and grassland plots in vegetated and bare plots but where are the bracken? Or is bracken vegetated? Please clarify which data were used when and why in a concise way.

- I think I miss an explanation why you chose to use temperature and soil moisture classes. It seems somewhat arbitrary at the moment.

- Figure 9: I understand the information in the figure and it seems a good explanation for the observed soil respiration patterns. Though, I assume the basis for this figure is the information on WTPT (Figure 4). But I don't understand how this information, which is based on WDPT tells you about flow paths. Please clarify. Comments to the Figures:

Figure 2 - This figure is really busy and the shadings in different directions and colours are overwhelming. Consider putting the rainfall data in a table. Or/and present the temperature curves as mean +/- standard deviation as they are following each other closely anyway.

Figure 3 - Suggest to move this graph to a supplement. I don't see a direct connection to the study other than that it presents the expected variation in soil moisture and soil temperature during the years.

Figure 4 - Where are your error bars on the barplots? I assume it is the means of

replicated samples?

Figure 5 - The presentation of soil respiration is challenging for the eye. Why not present boxplots? Picking out the means is very difficult in this way. Some sampling dates seem to miss the mean altogether. Same as with Figure 3, I think this information could go into a supplement.

Figures 6 following - You could combine figures 6, 7 and 8 to one figure with 3 panels. For figure 6, did you use data from bare and vegetated plots? How did you combine the soil respiration data? For figure 7, did you pool the forest and grassland data? Figure 8: how do you calculate the SWR distribution and what exactly does it mean? You mention it on page 8 line 8 but it is not clear to me how you calculate the distribution.

Figures 7 and 8: - Figure 7 shows soil moisture, Figure 8 shows SWR. To me, the information gained from both plots looks similar. What is the new information in Figure 8? I think I don't understand why you recommend the measurement of SWR (which is much more effort than SWC) if the same information can be gained from the measurement of SWC.

Specific comments: - Page2 line2: the reference to Karhu et al is wrong. Karhu et al themselves cite the reference that you need here. - Page 20 line17: where do you document the significant results mentioned? - Page 22 lines 18-20: what could the biological controls be? - Page 23 lines 12-14: reference to Figure 4: I can't related the information in this sentence to any information presented in Figure 4, please clarify. - Figure 9: the title of the Figure is misleading: The figure does not show soil CO2 efflux responses. It rather shows a theoretical framework of soil water distribution due to SWR.

―――――――――――――――――――

---

## Author Comment (AC1) · 23 May 2017

We are grateful to both reviewers for their detailed and constructive suggestions, which will allow us to further improve the manuscript. We are pleased to note that the reviewers share our view that the study is novel, interesting and timely. The questions raised on the relationship and respective roles of water repellency and soil moisture suggest that we have not made it sufficiently clear which effects we have directly determined and which are implied from the results and established knowledge about soil water repellency. This is an issue that we will clarify more specifically in the revised manuscript. We agree with most of the specific comments provided and will implement the suggested changes. The main issues raised are listed below marked with (R#1 or

[Figure]

R#2) and our responses on how they will be addressed in the revised manuscript are marked with (#A). There are a couple of suggestions that we do not fully agree with and hope we have given a sufficiently thorough explanation for our reasons.

Referee #1 (R#1) (R#1) The most important concern I have with the manuscript is that due to the strong co-correlation between soil temperature, soil water content and SWR it is not clearly distinguishable whether the observed effects on CO2 efflux were due to temperature/soil moisture or SWR.

(#A) The referee's concern about a strong co-correlation between soil temperature, soil water content and SWR and the difficulty to distinguish between the individual effects on soil respiration is fully justified. Indeed we therefore do not claim that water repellency itself controls soil CO2 fluxes. Instead, we suggest that SWR, by controlling soil moisture distribution, will affect soil respiration mainly in relation to heterotrophic respiration. The appearance and nature of soil water repellency is influenced by moisture and temperature, but once present, water repellency will strongly influence infiltration patters and resulting soil water distribution, which in turn affects respiration. It is this effect that is investigated in this study for the first time under field conditions. The finding that respiration is highest for patchy water repellency (within a confined temperature class) is clearly an important outcome that might not have been expected based on previous insights from laboratory studies.

(#R1) (...) SWR was determined only for the topsoil while soil respiration arises from the whole soil

(#A) Soil CO2 flux indeed results from the respiration over the full depth profile, however, previous studies (e.g. Fang and Moncrieff, 2005) have shown that the majority of soil respiration, especially heterotrophic respiration, originates from the top soil where the organic matter content is higher and consequently the carbon sources for microorganisms are high. Given that at both sites organic carbon contend below 10cm depth is very low it is reasonable to expect that the majority of soil respiration comes from the

top soil. Therefore focusing on soil moisture and SWR measurements within the top soil is sufficient for the purpose of this comparative study. We did measure soil water repellency at depth and on many occasions SWR was present up to 25cm depth, however, given their limited relevance for the study aims the results from the SWR depth distribution were not shown. Fang C, Moncrieff JB. 2005. The variation of soil microbial respiration with depth in relation to soil carbon composition. Plant and Soil, 268: 243-253. DOI: 10.1007/s11104-004-0278-4.

(#R1) There are several assumptions that are not justified based on the experimental findings of the study as well as inconsistencies in the discussion. It would certainly help to improve the manuscript if the results are treated and presented as being the outcome of a case study, meaning that a generalization of the observed effects is not necessarily possible.

(#A) We will go through the manuscript to improve consistency and make it clearer which findings are specific to this case study only and which can be reasonably expected to influence respiration in principle in soils affected water repellency elsewhere. Given that this is the first field study that examines the combined roles of water repellency, moisture and temperature in soil respiration, we feel it is important to the reader to highlight the potential wider implications of this case study. There is a substantial body of literature on water repellency and its effects on hydrology in soils from many regions around the world. From that it can be expected that the effect of water repellency on hydrological behaviour of most soils is fundamentally similar, but with the timing, duration and spatial extent of the effects being variable between different sites. In the revised manuscript we will make this clearer when discussing the results. This will include a statement that the magnitude of the hydrological effects on soil respiration will be site dependent therefore we suggest more studies to be done in the future to confirm the effect at other study sites.

(#R1) the title Title: The title states that spatially variable water repellency enhances soil respiration. This is not correct because it is not SWR itself but rather the (SWR-

affected) soil water content (and temperature) that actually controls soil respiration. Replacing 'enhances' by 'is associated with high' would therefore be more appropriate. Moreover, using the term 'spatially' in the title is somewhat misleading as it suggests that the study was focused on the spatial distribution of SWR at the study sites. However, deriving conclusions about the spatial distribution of SWR is simply not possible based on the investigation of only six soil cores per site.

(#A) As highlighted already above, we agree that water repellency does not DIRECTLY enhance water repellency, but respectfully disagree that it is incorrect to state that water repellency enhances respiration. A key outcome of the study is the evidence it provides for the ability of water repellency to enhance respiration through its effects on moisture distribution in the soil. In a similar vein, many studies have shown that e.g. obesity reduces life expectancy even so it is its indirect effects on blood pressure and diabetes (and their own first order effects) that reduce life expectancy.

We have indeed not determined the specific spatial distribution of water repellency. This could have only been done by destructive sampling, which is not possible in the context of repeated efflux measurements. We do, however, feel we have provided sufficient evidence for the presence of spatially variable water repellency and its influence on soil water distribution based on repeated water repellency and soil moisture measurements at the study site as a whole and an understanding of the effects of water repellency on soil water distribution from previous studies.

We therefore feel the title is justified and hope the hypotheses and supporting evidence provided in this study will be sufficient to trigger studies by other teams that will test the validity of our findings for other environments in future studies

P1L7: Here, hydrophobicity is used as a synonym of soil water repellency. This is not correct because SWR covers the entire range of states where soil repels water, while hydrophobicity explicitly denotes a state where water is not able to penetrate the soil (often defined as having a soil-water contact angle above 90 degrees)

(#A) The term hydrophobicity it often used synonymous with water repellency in the soil literature depending on the specific definition used. We agree, however, that it simplifies the text and will use water repellency throughout the body text.

P1L18: The authors discuss preferential flow as a possible mechanism to explain their results. This is fine in the main text, however, as this was not proved in the study it is conjecture and should not be in the abstract.

(#A) We will amend the abstract to emphasise the effect of water distribution patchiness rather than preferential flow being responsible for the higher respiration effect.

P4L6: What is meant by 20-m transect here? Is 20 m the distance between the plots on the left and the plots on the right? If yes, then including a scale would certainly help the reader because it is not immediately intelligible from Fig. 1 that the plots are arranged along a transect.

(#A) Thank you for pointing this out. The figure will be amended with a scale to make this clear.

P7L18-20: Given the total number of measurement events (n = 16) I was wondering whether the removal of soil material approx. 10 cm away from the flux collars would not influence the moisture distribution and hence $CO_2$ efflux. Could you please comment on that?

(#A) We would not expect that the removal of soil samples has affected the soil moisture condition, as the holes after soil removal were filled out with a similar material from the site to avoid such effects.

P8L7-8: The determination of WDPT frequency distribution and the SWR distribution parameter was based on measurements carried out on material from 4 depths at 6 plots. While SWR distribution with depth could be reasonably described, this is clearly not possible for the horizontal distribution as the plots were located several meters away from each other, not allowing to draw meaningful conclusions regarding the spatial

**BGD**

dependence and spatial structure of SWR. Moreover, considering that the material for the SWR determination was extracted at some distance from the flux collars, it seems very difficult to directly relate the measured $CO_2$ fluxes to the measured SWR distribution

(#A) As mentioned before, we intended to determine the specific spatial distribution of water repellency at each measurement event and correlate it with the soil $CO_2$ fluxes. This would ideally be done by destructive sampling, but this is not possible in the context of repeated efflux measurements. We therefore used the most viable alternative: repeated water repellency and soil moisture measurements at the study site as a whole. With these, and the established understanding of the effects of water repellency on soil water distribution from previous studies, we feel we have provided sufficient evidence for the presence of spatially variable water repellency and its influence on soil water distribution. The insights into SWR distribution are been based on 120 measurements per event and per site, which gives sufficient representation for SWR condition at the site.

P12L18: What is meant by 'surrounding'? As the plots are several meters away from each other, it is not possible to draw any conclusion about the conditions of the surrounding soil (i.e. in close proximity)

(#A) We agree that using the term 'surrounding' is not sufficiently specific and will change it to 'in close proximity'

P13, Figure 4: What is the rationale for using the standard error here (and in Figures 6, 7, 8 and Table 2)? Using the standard deviation (as in Table 1) is more appropriate to get an idea about the variation of the water content.

(#A) Standard error will be replaced by standard deviation in the Figures.

20L5: The authors assume that the SWR distribution parameter can be used as a proxy of heterogeneity in soil moisture distribution in the flux collars, however, the validity of this assumption was not proved in this study and seems highly questionable considering the points mentioned above.

(#A) SWR distribution presented in Fig 8 is a different presentation of results from Fig 4, which shows how variable the SWR was at each measurement event. SWR distribution was calculated from the percentage of the highest SWR persistence (>3600s) which represents the 'most extreme scenario' for SWR with expected lowest localised water contents and the thinnest water film on soil particles (according to Bachmann et al. 2008 and Derjaguin And Churaev, 1986, the more hydrophobic the soil, the thinner and more discontinued is the water film on soil particles). Soil with highest SWR distribution represents soils with similar SWR persistence for all investigated samples, while lower SWR distribution will represent soil of variable SWR persistence with patches of less and more water repellent and wettable soil. Based on the notion that higher SWR persistence will represent thinner and more discontinued water films we feel it is correct to use the SWR distribution as a proxy of heterogeneity of soil moisture distribution. We recognise that due to experimental constraints we can't refer the SWR distribution from adjacent soil sample directly with the soil flux collar therefore the combined results from all samples from each measurement event vs. mean $CO_2$ flux from all samples have been used to show how variable soil water contents can affect soil $CO_2$ fluxes in water repellent soils. We agree that the explanation given in this section of the manuscript were not sufficient and will therefore will amend the section to clarify better the rationale for calculating the SWR distribution and the meaning of it.

P20L8: The assumption that uniformly water repellent soil (SWR distribution = 1) is necessarily associated with homogeneously distributed low moisture content is not valid. This becomes immediately evident when considering that the calculation of this parameter is based on core material extracted from plots that were located several meters away from each other. Considering the dimension of the soil cores (5 cm diameter, 9 cm length) it becomes clear that the SWR distribution parameter is not representative of the site and not even representative of the individual plot. In other words, it is

easily conceivable that the wetting properties and thus the moisture distribution of the surrounding soil is different from that measured for the soil cores

(#A) This issue is already addressed in the previous comment. We reiterate that we feel that SWR distribution is the most effective way of giving a reasonable representation of the overall heterogeneity of water repellency for each sampling event (see explanation for 20L5). Given that it was based on 120 measurements for each event (6 sites, 4 depths and 5 measurements) we are confident it provided a sufficiently representative and statistically robust sample set to provide a reasonable overall estimate of heterogeneity of water repellency at of the sampling dates.

P22L21-22: Such detailed statements regarding SWR distribution at the sites are not justified (see comments above).

(#A) See comment above. Events where soil was exposed to long dry spells had indeed resulted in very consistent results with all showing high (WDPT>1hr) water repellency, in contrast to other events where the results where more variable. We will, however, remove the statement 'in the entire soil' as, indeed, we haven't measured the entire soil.

P23L3-5: Apart from the fact that spatial heterogeneity was actually not investigated in the present study (this is simply not possible by investigating only six soil cores per site) this statement is difficult to understand and in contrast to the assumption that SWR is the cause of preferential flow and a heterogeneous water distribution as stated, for instance, at P26L9-11. What is the authors' opinion? Is spatial variability of SWR caused by a spatially uneven infiltration into the soil which, in turn, is affected by preferential flow, or is SWR itself the cause of an uneven water infiltration and preferential flow phenomena?

(#A) As explained before we intended to measure the spatial heterogeneity of SWR at each site and relate that to soil $CO_2$ fluxes. As it has been shown in many different studies, SWR causes the uneven infiltration after dry spells, enhanced preferential flow

and can cause very patchy soil water distribution. We will clarify the message in the revised manuscript to avoid any confusion.

P23L10-12: The statement in this sentence is not clear (see comment above). It is not proper to state that the preferential flow paths caused by SWR resulted in a high spatial variability of SWR.

(#A) The work done previously on water repellent soil with presence of roots and stones (cited in the manuscript) has shown that water infiltrates into the macropores created by the 'obstacles' to move downwards leaving majority of the soil water repellent and only near the 'obstacles' switching of wettability takes place in a progressive way. Soil water distribution expands towards the soil matrix away from the preferential flow paths and wetting of more soil takes place. This could indeed not be monitored directly in the current study. However, based on previous work, this can reasonably be expected to take place in water repellent soils under field conditions. There is no reason to assume that our field sites would be exceptions from this behaviour.

P24L18-20: The statement in this sentence (high CO2 flux at high water content) is in contrast to the findings presented in Figure 7 and the conclusions and are not consistent with the 'model' resented.

(#A) Thank you for spotting this. That was a mistake indeed and it will be corrected accordingly.

P24L25: What is meant by 'severity of SWR'? Is it different from 'persistence of SWR'?

(#A) In this sentence we refer to the work of Goebel et al and Lemparter et al. who have measured soil water repellency by determining the contact angle of the soil. The measurement of contact angle between the soil and the liquid gives an indication of severity of SWR rather than water infiltration persistence. Several studies showed relatively good correlation between the severity and the persistence of SWR in different soils, but it is important to refer to the work methodology using the correct terminology.

P25L8-10: The use of 'response' is not justified in this context because it is not SWR itself but rather the SWC (influenced by SWR) that actually influences soil respiration. Using 'associated' would be more appropriate ('... different CO2 fluxes were associated with different patterns of SWR ...').

(#A) We agree. The sentence will be corrected as suggested.

P25L11: Please check this sentence. What is meant by '... the more realistic effect of SWR ...'? (more realistic than ... ?).

(#A) The sentence will be corrected. Instead of 'realistic effect' we will use 'representative'

P25L12: I have some issues with the 'conceptual model' presented in Figure 9. According to the model, wettable soil (Figure 9a) represents a condition where soil moisture is too high or soil temperature is too low for SWR to develop. The CO2 efflux associated with this particular state was found to be low. However, it was not SWR that caused the low CO2 efflux but rather the high water contents or the low temperatures (as was correctly stated by the authors). Hence, it is not justified to state that the model is accounting for the complex effect of SWR as both SWR and CO2 efflux are simply co-correlated and controlled by soil moisture and soil temperature. In addition, Figure 9c, which represents the 'water repellent state' with uniformly water repellent soil suggests extremely low water contents (near zero) as compared to the other states. Apart from the general problem of relating the measured parameters in the present study (please see comment to P20L8), the results presented in Figure 4 show that this is not necessarily the case. As shown in Figure 4a, there was a transition from a uniformly water repellent soil (on 19/7/13) to a variably water repellent soil (on 29/8/13 and 8/10/13), while the corresponding water content remained fairly constant around 10 vol-%, which is far from being completely dry (as suggested in Figure 9c). There is also some ambiguity about the intermediate (variably water repellent) state illustrated in Fig. 9b. What do the authors really think? Is SWR the cause of an uneven water infiltration and

causes preferential flow phenomena, or is it the spatially uneven infiltration into the soil which, in turn, is affected by preferential flow that causes the high spatial variability of SWR (as stated at P23L3-5)? Generally, the proposed 'model' would only be valid for the specific conditions of the sites investigated. For instance, it is well conceivable that a wettable soil is characterized by an intermediate water content (particularly in case of sandy soils). And the occurrence of such a situation is also possible in summer as shown in a study by Buczko et al. (2007, Ecological Engineering 31: 154–164). Under such conditions (i.e. intermediate water contents and high temperature) microbial activity and $CO_2$ efflux can be expected to be high (and might be even higher than for variably water repellent soil). Overall, given the lack in general validity and explanatory power, using the term 'model' seems not appropriate, although the given explanations and the illustrations in Fig. 9 are valuable for understanding the observed effects on $CO_2$ efflux at the investigated sites.

(#A) We have attempted to explain the concept of different hydrological conditions caused by presence of SWR and how this can affect soil respiration. In this concept (model) we are not representing the soils that are continuously wettable independent of the temperature and the moisture status, but soil which will turn water repellent when exposed to low soil moisture contents, usually also related to higher soil temperatures. In the model we show that soils prone to development of SWR will be wettable only when soil moisture is high and the temperatures are low and therefore it will be associated with low respiration rates, resulting from the temperature and high soil moisture effect. It is indeed the temperature and moisture effect on soil respiration rather than soil wettability on its own. We will clarify this paragraph to show the message more clearly. We will also amend the text and the Fig 9c graph to show that some residual water content can be present although it will be low and the connectivity between the pores will be severely disrupted which will result in low respiration rates. We agree with the reviewer that after frequent rainfall soils can become wettable during the summer at high temperatures (similar to Buczko at al. study). At the site it was observed especially during the 2014 summer where majority of soil was wettable, but despite high

temperatures soil respiration was low see Fig5 & Fig4 for grassland. We understand that the findings of Buczko may suggest the respiration under hot and not moist conditions can be expected to be high, but the respiration rates have not been measured in that study, so it is the referee's speculations rather than demonstrated behaviour that the respiration rates were high in that case.

P26L16-19: Again, it is not reasonable to state that the intermediate state of SWR enhances soil respiration. It is indeed conceivable that $CO_2$ efflux of a wettable soil, which is characterized by an intermediate and homogeneously distributed water content, is even higher than of a variable water repellent soil, provided that the temperature is high enough (see comment above and comments to P25L8-10 and the title)

(#A) We would like to stress again that we were studying soil prone to development of soil water repellency, which below a certain soil moisture content & above a certain temperature will become water repellent. We don't claim that variably water repellent soils will have higher respiration rates than wettable soil at the same moisture level, this is indeed not possible to examine with the current research design. We will amend the text to clarify the issue raised by the referee.

P29L10-19: The conclusions presented here are not justified (see comments above). (#A)The text will be amended accordingly with changes in the discussion.

(#A) All minor issues listed by the reviewer below will be addressed as suggested.

Other minor points: P1L12: SWR is introduced at P1L7 and should subsequently be used instead of 'soil water repellency' throughout the text. This should be checked carefully as there are many instances where 'soil water repellency' or 'water repellency' is used. P2L5-7: The statement that soil moisture controls pore-water connectivity is self-evident and should be removed. P3L4: SOC is introduced at P2L6 and should subsequently be used instead of 'soil organic C' throughout the text. P3L18: Please check the style of the sentence (..., which ....., which). P4L8: Please replace 'for' by 'at' (At each study site ..., and at each ...) P6, Table 1: Please replace 'for' by 'of' (Selected

soil properties of samples ...) P7L6: Please replace 'for' by 'at' (At each study plot ...) and 'was' by 'were' (... soil collar were temporarily removed ...). P7L12: I would suggest to replace the sentence by: '... was determined by fitting an exponential function to the evolution of $CO_2$ concentration over time ...'. P7L13-14: Please check the style of this sentence. In addition, could you please add information about the overall percentage of fittings with $R^2 < 0.95$. P7L19: Can you please specify what is meant by 'further soil measurements'. P8L3: Was bulk density really determined after each field visit? P8L7-8: Could you please state how many replicate measurements per plot and depth were carried out. P8L18: Please check this sentence ('... to reduce oxides of N, $CO_2$ and $N_2$ were determined...') P8L20: 'distilled' or rather 'deionized' water? P8L22-23: Please use SWC instead of 'soil water content'. This should be checked carefully throughout the text. P9L2-3: Could you please state the post-hoc test used in conjunction with the ANOVA. P10, Figure 2: Please replace 'Air Temp' by 'Air temperature'. P11, Figure 3: Consistent labeling should be used ('Soil temperature', 'vol-%'). Please use either 'Sampling event' or 'Soil sampling' in the legend. Is it correct that Fig. 3a begins with June 2013 while Fig. 3b begins with July 2013? P12L6-7: Please use the same rank order for text and numbers (from low to high), i.e., '... slight to moderate (WDPT 6 to 600 s) ...' and '... slight to extreme SWR (WDPT 6 to >3600 s). P13, Figure 4: Please be consistent with the labeling used in Figure 3 (vol-%) and use the same labeling for a and b (either 'Soil sample collection date' or 'Sample collection date'). Please use site designations consistently throughout the text and figures. Cur- rently there are several variants, e.g., forest (T-f), forest site (T-f), Thetford-forest (T-f), etc. P14L10: Is 14âŮę C correct? Figure 6 shows the highest fluxes at the forest site to be around 16âŮę C. Is there any explanation for the large difference in temperature where the maximum $CO_2$ fluxes were found? P17, Figure 7: Please insert '(âŮęC)' after 'Soil temp.'. 'temperature ranges' -> 'tempera- ture bands'. Please replace '... for SWC's grouped into 10% SWC ...' by '... for SWC grouped into classes of 10 vol-% ...'. P18, Table 2: The case '\*\*p<0.01' does not appear in the table and should be removed. P19, Table 3: Using \* for referring to the footnote is not appropriate here as \* is also

**BGD**

used in the interaction term 'SWC * Temp'. P19L18-21: This paragraph is not adequate in the Results section and should be moved to the Discussion. P21, Figure 8: Please insert '(âŮȩC)' after 'Soil temp.'. Please use a consistent description of the temperature bands in Figure 8 and Figure 7 (P17). P22L6 and L15: These statements here are inconsistent ('SWR was present for most of spring, summer and autumn' vs. 'SWR was observed from early summer until late autumn'). P22L23: Please delete 'and' in this sentence to read: '... frequent change between sufficiently dry and wet periods, ...'. P22L24: Please change to '... which allows development ...'. P23L3: Please replace the comma by 'and' to read: '... higher than 2013 and 20% higher than 2015.'. P23L19: What is meant by C fluxes here? Referring to soil respiration would be suf- ficient here as no other C fluxes (e.g. transport of dissolved organic matter) were investigated in the present study. P24L13: The reference is lacking: what is meant by 'this forest type'? This needs to be specified. P24L16: Using 'but' in the context of this sentence is not appropriate. P25L5-7: Please check the style of this sentence ('... wettability conditions with uni- formly low (wettable) and high (extreme) water repellency ...' as well as '... when soil is dominated either by wettable soil ...'). P25L12: Please check this sentence ('Wettable soil ... represents a condition observed when a soil water repellency is absent...'). P29L6: Please add an 's' to read '... becomes severely ...'. P29L10: Please change to '... were indeed associated ...'.

---

## Author Comment (AC2) · 23 May 2017

We are grateful to both reviewers for their detailed and constructive suggestions, which will allow us to further improve the manuscript. We are pleased to note that the reviewers share our view that the study is novel, interesting and timely. The questions raised on the relationship and respective roles of water repellency and soil moisture suggest that we have not made it sufficiently clear which effects we have directly determined and which are implied from the results and established knowledge about soil water repellency. This is an issue that we will clarify more specifically in the revised manuscript. We agree with most of the specific comments provided and will implement the suggested changes. The main issues raised are listed below marked with (R#1 or

R#2) and our responses on how they will be addressed in the revised manuscript are marked with (#A). There are a couple of suggestions that we do not fully agree with and hope we have given a sufficiently thorough explanation for our reasons.

Response to Referee #2

The authors present a lot of data, which I find overwhelming. In my opinion, the manuscript would benefit from focusing, especially of the results section. Data that are not crucial for the explanation could go into supplementary information. This would guaranty that it is not an information overflow.

(#A) The changes suggested by Referee 1 already go some to focus the manuscript and we will revise the text to ensure further tightening. We do, however, feel that the figures included in the manuscript are necessary to understand the complexity of the phenomenon. It is already clear from Reviewer 1's comments that we have not made it sufficiently clear what is demonstrated by the data and what is inferred. The figures assist in making this clearer. Given that Referee 1 appears happy with the number of figures and has suggested some alterations to them, indicates the value of maintaining the figures in the main text.

The experimental setup as described in Figure 1 and Table 1: I have a hard time following why bracken and vegetated soil was measured, and also what the information on bare soil and vegetation soil was used for. E.g. Figure 4 presents data from which plots exactly? And Figure 5 displays forest and grassland plots in vegetated and bare plots but where are the bracken? Or is bracken vegetated? Please clarify which data were used when and why in a concise way.

(#A) We appreciate that the plot vegetation and surface bareness could be slightly confusing. First of all we have 2 sites: grassland and forest. Both sites have variable vegetation type cover (bracken and grass) which was thought important to be included in the study design given that the vegetation type may affect development of soil water repellency as well as soil respiration differently. As it can be seen on Figure1, 6 study

plots have been established at each study site, plots with bracken and grass vegetation cover. To differentiate the CO2 flux origin from accumulated litter and soil only, vegetation has been removed from one collar of each plot. The differences in SWR between bracken and grass were insignificant, therefore the results from all plots were analyzed jointly. After CO2 flux measurement for bare soil, the soil cover has been put back on the soil surface to allow litter leaching into the soil and reduce enhanced drying of the soil. Given that the soil samples were collected from vegetated part of the plot we were able to correlate CO2 flux and SWR distribution only from vegetated plots. We will clarify the text and the figures to better explain why soil under different vegetation were used for the study and which results represent joint and which the separate results from study plots.

- I think I miss an explanation why you chose to use temperature and soil moisture classes. It seems somewhat arbitrary at the moment.

(#A) Given that the measurements were conducted under field conditions the results of soil moisture and temperature had similar, but not exactly the same values it was necessary to group the results into the moisture and temperature classes. As with most classifications of environmental values, they are indeed essentially arbitrary, but facilitate comparison and interpretation of complex datasets. We will clarify this point in the revised manuscript.

- Figure 9: I understand the information in the figure and it seems a good explanation for the observed soil respiration patterns. Though, I assume the basis for this figure is the information on WTPT (Figure 4). But I don't understand how this information, which is based on WDPT tells you about flow paths. Please clarify

(#A) We like your suggestion to entitle the figure 'Theoretical framework of soil water distribution due to SWR and its effects on soil CO2 fluxes'. Referee#1 has also raised some questions about the conceptual model therefore we plan to amend this section and the figure to explain our understanding of effects of variable SWR and soil moisture

distribution on soil CO2 flux. As pointed out by the Ref#1 we will refer to soil water distribution rather than preferential flow in the 9b figure and we aim to amend the figure accordingly. We will also make some corrections to 9c figure and description to better visualize the water films and soil moisture content in soil pores.

Figure 2 - This figure is really busy and the shadings in different directions and colours are overwhelming. Consider putting the rainfall data in a table. Or/and present the temperature curves as mean +/- standard deviation as they are following each other closely anyway.

(#A) The business of the figure seems to be arising mainly from different patterns used. We wanted to make the figure grayscale printer friendly. We will revise the figure using colour, convert it into a table, or present two separate figures in the supplementary material as suggested.

Figure 3 - Suggest to move this graph to a supplement. I don't see a direct connection to the study other than that it presents the expected variation in soil moisture and soil temperature during the years.

(#A) We feel this figure is important and should remain in the main text as it shows important information for the reader about the measurement dates and puts them into a context with the moisture and temperature data. We will clarify this point in the text.

Figure 4 - Where are your error bars on the barplots? I assume it is the means of replicated samples?

(#A) Figure 4 represents the frequency distribution of all individual samples and replicates (not only averages) and therefore the use of the error bars is not applicable. The variability of the results between the samples is represented by different colors of the bar. This will be clarified in the caption.

Figure 5 - The presentation of soil respiration is challenging for the eye. Why not present boxplots? Picking out the means is very difficult in this way. Some sampling

dates seem to miss the mean altogether. Same as with Figure 3, I think this information could go into a supplement.

(#A) We also feel this figure is important in and should remain, however, we would be happy to make the suggested changes to the graph and to present the mean values more clearly and clarify its relevance more specifically in the main text.

Figures 6 following - You could combine figures 6, 7 and 8 to one figure with 3 panels. For figure 6, did you use data from bare and vegetated plots? How did you combine the soil respiration data? For figure 7, did you pool the forest and grassland data? Figure 8: how do you calculate the SWR distribution and what exactly does it mean? You mention it on page 8 line 8 but it is not clear to me how you calculate the distribution.

(#A) Thank you for the suggestion to combine the 3 figures into one graph panel. We agree that this it is a good idea and will combine it in the revised manuscript. We will also clarify which data have been combined and how the SWR distribution was calculated in the methodology section.

Figures 7 and 8: - Figure 7 shows soil moisture, Figure 8 shows SWR. To me, the information gained from both plots looks similar. What is the new information in Figure 8? I think I don't understand why you recommend the measurement of SWR (which is much more effort than SWC) if the same information can be gained from the measurement of SWC.

(#A) Soil moisture is expected to be very variable in soils with variable soil water repellency and that could cause a different heterotrophic respiration due to the patchiness of soil moisture distribution. Moisture content is clearly a key driver, but given that trends associated with climatic changes may lead to increased severity of soil water repellency, it is necessary to understand more fully what effect it will have on soil respiration. We feel this study makes an important first step in that direction, however, more studies in which water repellency is measured are needed to determine to what degree the patterns and implications highlighted in this study are applicable elsewhere.

(#A) The remaining and more specific suggested edits below will be addressed as recommended by the referee

Specific comments: - Page2 line2: the reference to Karhu et al is wrong. Karhu et al themselves cite the reference that you need here. Page 20 line17: where do you document the significant results mentioned? Page 22 lines 18-20: what could the biological controls be? Page 23 lines 12-14: reference to Figure 4: I can't relate the information in this sentence to any information presented in Figure 4, please clarify. Figure 9: the title of the Figure is misleading: The figure does not show soil $CO_2$ efflux responses. It rather shows a theoretical framework of soil water distribution due to SWR.

---

## Referee Report (RR1)

With the responses to the referees, the authors have greatly improved the manuscript. The only issue that has, in my opinion, not been sufficiently addressed is the display of Fig. 4. I still find it very hard to see which dots belong to bare soil and which ones belong to vegetated soils. This makes it a bit hard to believe the statistics presented in table 2. Additionally, the grammar of the manuscript needs improvement (please see comments below).

Title: now to me is a bit meaningless as it does not state what to expect, although it is correct.

P1L8: "causes reduced" -> reduces
P1L10: add commas: dynamics, and specifically on CO2 efflux, have
P1L14: Soil CO2 efflux. We conducted in situ field based measurements which were carried out… (though, in situ could be deleted as field measurements are already in situ)
P1L19: suggest to replace diminished -> reduced
P1L21/22: … with different characteristics related to CO2 production and transport.

P2L18: 2016), but also (insert comma)
P2L21: Why not say "Most soils are very …"

P3L3: "offered" seems the wrong word. What about "created by"?
P3L4/5: sentence grammar needs correcting: "Irregular water infiltration in water-repellent soil often creates distinct zones with water filled pores, concentrated dissolved organic C and …" (remove comma, remove "a", use C instead of carbon"

P3L8: hydrophobic particle-surfaces; or is it possible just to say hydrophobic surfaces?
P3L9: simplify sentence: "has been reported to reduce soil microbial…"
P3L11: food and water sources (plural)
P3L13: soil organic matter (SOM)
P3L14: add comma: "water-repellent state, and…"
P3L18: soil respiration (i.e. CO2 efflux) – I think you should either introduce this before (e.g. when you first use the term soil respiration, or delete "(i.e. CO2 efflux)"
P3L19: remove however
P3L20/21: simplify to: "The aim of the current study is to investigate soil CO2 efflux responses…". To the reader it is already clear that the novelty of the study is that you explore something that has not yet been done. You emphasise this with the paragraph before. No reason to overdo it (In my opinion).
Also, as mentioned before, in-situ is field. In line P1L14 you write in-situ without hyphen -> decide which one to use if you feel the word is necessary.
P3L22: consider changing "real world" to "natural"

P4L4: six study plots
P4L6: When is the growing season? Could you add this information in brackets? -> e.g. growing seasons (Apr-Aug)
P4L7: Suggest splitting the sentence: "… years (2013-2015). SWC, temperature, CO2 efflux and SWR was measured in approximately monthly intervals."
P4L9: change vegetation plot -> study plot (to be the same as in the figure caption. Additionally, it is clearer because you have two vegetation types.
P4L9-11: for clarity consider the following:

Both study sites consisted of six plots with two PVC collars for CO2 efflux measurements (n=12) arranged along a 20-m transect (Fig. 1). Grass and bracken vegetation was covered equally. At each study plot, soil respiration was measured on vegetated soil and on bare soil respectively. Bare soil measurements were conducted on soil collars from which the vegetation and litter layer inside the collar was temporarily removed for the duration of the $CO_2$ efflux measurement to assess the contribution of different layers to total soil respiration, and put back after the measurement. The sites were monitored during the growing seasons in three consecutive years (2013-2015), involving continuous measurement of SWC and soil temperature, and recording of $CO_2$ efflux and persistence of SWR during site visits at approximately monthly intervals.

Also, you repeat part of this information on P8. Chose one place and remove in the other.

P4L15: "C and N" – you have not yet specified that N = nitrogen

P5L2: sites (plural)
P5L3: 20-m (insert hyphen as in text)
P5L5: from your explanation to the referee comments, it is clear what you did. I suggest to change "bare soil with vegetation temporarily removed" to "vegetation temporarily removed for soil CO2 efflux measurements = bare soil"

P6L2/3: it is not obvious to me why the abbreviation of the sites need a "T" – why not call it Grass and Forest instead? Which would make distinguishing the sites when reading the paper much clearer.
P6L6: since 1995 -> I assume the network was established in 1995 and is running since 1995. Choses either or.
P6L9: "site was converted"
P6L10: for simplification consider: "The dominant vegetation cover and soil level was similar for both sites with large areas…"
P6L12: "was also present"

P7: Add to the Table header that T-f is forest and T-g is grassland. I'd suggest to move the information "(mean (st.dev))" that is currently in the table to the Figure header. Although you say see main text for more information, it seems useful to know that the SD for each was calculated for n=6

P8L2: information in brackets could be removed i.e. (twelve per study site)
P8L2: For simplification consider: … were inserted into the soil for CO2 efflux measurements.
P8L6-8: this is a nice description, but repeated (P4L8-12) – I personally find this description better.
P8L9: CO2 efflux was…
P812/13: "… exponential function to the accumulation of CO2 over time..."
P8L14: was below 0.95"
P8L16 "5 cm deep"
P8L16/17: … at 5 and 10 cm depth was conducted at both study sites/all plots…" – something seems wrong with the sentence, please check. Please specify if measurements were conducted at each plot (n=6 per site) or at each site (n=2 in total)

P9L2: "at constant temperature (xx oC) for 24 hr"
P9L4: add what WDPT means
P9L7: should <5 be <6?
P9L10: remove "essentially"
P9L12: "diverse soil water distributions." (plural)
P9L15: N – should be defined earlier
P9L18: "using gas chromatography" – remove the

P9L18: I assume that C:N ratios were not calculated based on peak areas but based on elemental composition?

P9L23: "narrow range (e.g. …)" – remove values

P10L2: removed analysed

P10L3: ANOVA and post hoc test were used (plural)

P10L10: summers (plural)

P10L13: temperatures (plural)

P10L14 suggest to remove "environment"

P10L15: suggest: … at 5 cm depth at the T-g and T-f site, respectively

P10L18: water content in the top soil… than lower down in the soil profile, while…

P11L4 (Figure heading) you mention soil temperature before moisture although it is located on the second axis. It would be more intuitive to change it around

P11L5: this is the first time that you call it "Thetford-forest" (and grassland) – consider introducing it in the method section.

P12L2: Suggestion to change the structure slightly and split the sentence into two: "At least some degree of SWR occurred during the summer months… followed by increased soil wettability in the colder… at both sites (Fig. 3). However, SWR patterns varied from…"

P12L4: 2015, when (insert comma)

P12L5: WDPT > 6 : would this be above 5? When looking at Fig. 3, the green colour has a range of 6-10.

P12L6: while the other seasons

P12L11: exhibiting the full range

P12L12: the WDPTs corresponded (it is times, right?, there are no time values) (also in P12L16); similar in P12L13 – suggest to call it SWCs

P12L16: Thus, soil at the T-f site… (spelling it out makes it easier to follow)

P12L17: occurrences (plural)

P12L17: samples remaining water-repellent: do you mean "being water repellent? The word "remaining" seems wrong. Please check that the sentence says what you want to say.

P13 Figure caption: please spell out SWR and WDPT. Is it really SWR persistence or is it SWR? Suggestion: SWR measured by water drop penetration time (WDPT) for the topsoil 0-9 cm. Frequency distribution of WDPT ranges for 120 measurements per sampling date."; in Figure: blue range should be <6?

P14L3: lowest $CO_2$ effluxes were … (plural); same in P14L4

P14L4: suggest: variability in $CO_2$ efflux rates between samples.

P14L5/6: Bare soil plots showed signi. lower $CO_2$ efflux… - I cannot see this in the plot but the statistic is clear. However, I think I would rephrase that sentence to saying that vegetation plots in T-f showed significantly higher $CO_2$ efflux than bare soil. My line of thinking is that vegetation will add to $CO_2$ efflux by its presence and bare soil is the baseline.

P14L8: Why is it 10 or 12 oC? Is it for each of the sites?

P14L10: "…grassland site (T=g), a reduction in $CO_2$…"  - add "site" and remove "however"; Also, define what is "former".

P14L12: total variation in (singular)

P14L12: why not just saying: "by considering soil temperature and moisture together…"

P14L14: and can lead to

P14L15: reduced again at high SWCs. (not SWC values)

P14L15: what is a very limited effect? Remove "very"

P14L18: improved the explanation

P15 Figure legend: vegetated (filled circles, grass and bracken plots combined, n=6); The explanation at the end "with both bracken and grass plots" added at the end is rather confusing. In general, to me, the visibility of information shown in the figure has not improved from the last version. Figure 5a shows the point you want to make (summer-higher temperatures-higher CO2, winter-lower temperatures-less CO2 efflux) much better in my opinion.

P16: say that SWC is soil water content and SWR is soil water repellency. Though, is SWR not the frequency distribution of WDPT ranges? (You explain what you did in P19L4/5 – it might be worth adding this to the Figure or add to the methods section) P15L5: SWC – move up to L2; L5 and L7: change fluxes to effluxes.

P18 Table header: CO2 effluxes

P19L1: section 3.2 uses SWR – be consistent
P19L2/3: "and at the site it was observed at higher soil temperatures and lower SWC" – I don't understand what you want to say. Please clarify.
P19L6: represents soil (use present past), same for denoted -> denotes; also next line
P19L12: around -> between
P19L15: consider: events -> dates/time points

P20L6: grassland revealed (remove comma)
P20L11: considered to be a state most susceptible…
P20L17: consider: pinpointed -> determined
P20L17: "in this study within each year" - remove "within each year" as it is not relevant as it was not determined in any of the years.
P20L20: entire warmer periods – suggest to remove "entire"
P20L21: soil at the site – do you mean sites (plural) or one of the sites (if so, specify which one)?
P20L25: throughout what? Suggest to remove "throughout"
P20L26: soil areas – what are the areas? Do you mean between plots? Or depths?

P21L3: and the partial
P21L3/4: was likely a consequence of
P21L3-5: sentence starting with "The high spatial variability…" The sentence does not makes sense to me. Please check and it says what you want to say.
P21L7: anticipate: I think it's the wrong word. Consider believe/assume/suppose or similar
P21L10: had a lower soil (insert a)
P21L11: lower water-repellent soil was moist
P21L13: have been not only deficient (remove one been)
P21L15: and dissolved organic C is expected in wetter zones.
P21L19: be also responsible -> remove also
P21L19: soil respiration and C fluxes – which other C flux then soil respiration have you measured? Remove one of the terms.
P21L20: Thus, no (add comma) – but better use: Therefore, no…
P21L22: than measurements during the warmer (add measurements)
P21L24: seasonal fluctuation of CO2 efflux - ? if so, please add, if not, then please add
P21L25: "it is clear that the latter constitute the main factor affecting soil respiration" – firstly, it should say "constitutes", secondly, it is not "clear" but it is likely that temperature drives soil respiration to a certain level. Please rephrase.

P22L5/6: specify if a maximum level of CO2 efflux or temperature was reached;
P22L6: when soil CO2 efflux was (singular) or CO2 effluxes were (plural)
P22L8: was the restricting factor for soil CO2 efflux.
P22L9: measurements of low,…
P22L11: (Or et al. 2007), and (insert comma)
P22L12: cause of a decrease in CO2 efflux, primarily
P22L14: soil CO2 efflux (singular) – until now, you always used the singular
P22L15: efflux was (singular)
P22L15: changing SWC, particularly at high soil temperatures (Fig. 5b), and the …
P22L16: high, especially at intermediate SWCs.
P22L18: variable, most
P22L18/19: in a heterogeneous soil moisture distribution. – what is "very heterogeneous, there is a maximum; also, it should be a heterogeneous distributions or heterogeneous distributions)
P22L19: to the development
P22L20: by the presence
P22L25: wide range of scenarios – scenarios of what? I assume ranges of SWR? But its not really clear

P23L1: WDPts … not allow the identification of
P23L3: and the proportion
P23L5: your (a), (b) and (c) would be better used if they were the same as in Fig. 6.
P23L5/6: suggestion: when soil is neither dominated by wettable nor water repellent soil patches…
P23L10: when, due to frequent rainfall, SWR disappeared
P23L10: occasions, low
P23L13: SWR distributions
P23L14: and soil moisture was low.
P23L14: suggest to delete "In the latter case" – the sentence is easier to understand without it
P23L16: similar highly water-repellent soil
P23L17: Owing to -> Due to
P23L18: it ceases -> activity ceases
P23L21: and, this thickness

P24L2: suggest: in the UK in the future
P24L3: during the rewetting
P24L4: Muhr et al who observed a slower – deleted "observed" later in the sentence; that could have been caused by SWR
P24L5: distributions (plural)
P24L5/6: when SWR, and … distribution, was (add commas)
P24L7: which creates
P24L8: "have" – suggest to exchange for provide/contain
P24L8: paths, are
P24L11/12: and the atmosphere
P24L13: this is likely (add is) – but, what's the name of the most common soil condition – replace this by… I guess variably water-repellent soils" (?)
P24L16: suggest to remove "therefore"
P24L17: suggest to replace real world -> natural, field conditions
P24L18: suggest to remove "and its effects are clearly more complex as discussed below" – below is not a great deal of discussion that would (in my opinion) justify this part of the sentence. You more related to points you have already discussed above.
P24L21: should it be particles (plural)?
P24L24: Fig. 6

P24L24: "sufficiently simple to be fundamentally applicable" – it is correct but really hard to read. Consider something simpler. (I would remove the word "fundamentally").

P25L2: this scenario: replace with "SWR"

P27L2: soil respiration, and
P27L3: lead to (singular)
P27L4: exhibit -> suggest to use showed (past tense in any case)
P27L4: which are also (plural)
P27L8: water-repellent soils.
P27L10: remove throughout, or make clear what it relates to

P27L10: soil CO2 effluxes (plural)
P27L11: SWR distribution resulted (remove comma and use past tense)
P27L14: low, or when there was; there was a high; in general, reconsider sentence starting "A wettable soil state…" something is not quite right.
P27L16: suggest to use "natural conditions instead of real world field conditions
P27L16: suggest to remove "examined for the first time here" – you already say that at the beginning of the conclusions
P27L19: soil zones, it can actually
P27L20: SWR, measurements should therefore not…
P27L20: this sentence misses a but – basically it misses your suggestion – please clarify – you probably want to combine it with the following sentence
P27L22: allow the prediction of responses of
P27L24: predictions, and (add comma)
P27L24: sentence starting with "In view of…" – it's your last sentence but it is so long that it is difficult to extract the core message of it. Suggest to split it up to make your last message easily accessible to the reader.

P28L2: foundation for the prediction of C dynamics under

---

## Author Response (AR2)

Response to Referee comments.

We are grateful to both reviewers for their detailed and constructive suggestions, which will allow us to further improve the manuscript. We are pleased to know that the reviewers felt that the manuscript quality has been greatly improved and only minor corrections were needed.

All of the detailed suggestions made by the reviewers, mostly related to English grammar or style, were corrected as suggested by the referees. We have addressed all the comments were further clarification was requested. There were only a very few minor suggestions which we didn't implement, but instead we have rewritten the sentences to improve the clarification of the text.

We really hope at the current stage the manuscript will be accepted for the publication.

Kind regards,

Emilia Urbanek

**Ref1**

All referee 1 suggestions have been implemented unless suggestions from Ref#2 were used instead.

**Ref 2**

All marks highlighted yellow have been implemented as suggested. Green highlighted comments have been addressed differently than suggested by the reviewer

P1L7: CO2 should be introduced as this was done for C in L10.

P1L23: Please check this passage: ' ... higher nutrient concentration for microbial activity resulting in high CO2...'.

P2L7: O2 should be introduced.

P2L21: Please check the order of references given (please carefully check the entire text; e.g. P3L5-6, P23L18).

P2L25: Please delete ';' after Ritsema & Dekker, 2000.

P3L3: please use 'C' instead of 'carbon' here (as introduced at P2L13).

P3L13: please define 'SOM'.

P5L4: it should read 'filled circles' instead of 'closed circles'.

P8L13-14: Please check this passage: '... where the function fitting R2 was below value of 0.95...'.

P9L3: please replace '-' by '.' to read: '... 6.7-9.2 cm ...'.

P9L3: Bulk density was deleted from this sentence (presumably because it wasn't measured after each field visit), however, as it is presented in Table 1 and was necessary to convert gravimetric into volumetric water content its determination should be mentioned in the text.

P9L6: 'were' should be replaced by 'was'.

P9L18: Is measurement via GC correct? Or were C and N rather measured by IRMS after separation on a GC column? Please check this. The methodology is correct as stated in the text

P9L23: please insert 'of' between 'range' and 'values' and delete temp in this sentence.

P9L24: why '6.1-10'? This would not include values >6.0 and <6.1. Hence, '6-10' would be more adequate.

P10L2: Could you please elaborate a little bit more on the linear mixed model procedure you used. Why did you use a mixed model and not a simple linear model. Usually, a mixed model is used to include random-effects factors. So which factors did you define as random-effects factors in your analysis? Additional information has been added to the text

P10L3: '... the ANOVA test ...' sounds somewhat strange. Writing '... comparison, ANOVA and Tukey's post hoc test ...' would be better.

P11: Please check the alignment and the scaling of the figure (10/2015 vs. 11/2015).

P12L11: please replace 'a full range' by 'the full range'.

P13: please add information about n (e.g. n = 24) as was given in the original manuscript.

P14L12: please use the singular form: 'variation' instead of 'variations'.

P15: please check the scaling (11/2015 vs. 12/2015) and the alignment.

P16: Caption to Fig. 5: SWC and SWR should be written in full at their first appearance and afterwards abbreviated. Please add information about n. Why using SWC's or SWCs? SWC may stand for both singular and plural. It should be 'temperature classes' instead of 'temperatures classes'. Please add +/- in L6 (i.e. +/- 2°C). It should be '+/- 0.1' instead of '+/-0.2' (grouping of SWR distribution).

P17: Why reporting on SE instead of SD (as given in the figs and Table 1)? Again, please add information about n.

P19L2: Please could you explain what is meant by stating '... it was observed at higher soil temperatures and lower SWC ...'.

P19L4: 'were converted' instead of 'was converted'.

P19L5: please use either '0' and '1' or 'zero' and 'one'.

P19L10: please be more modest here: 'association' instead of 'response' would be more appropriate.

P19L11: 'flux' -> 'efflux'.

P20L6: please check consistency: early autumn vs. late autumn (as written in L15).

P20L10-11: please add 'a' between 'be' and 'situation'.

P20L16-18: please check the reference in this sentence. The statement '... clearly associated with low moisture contents and higher soil temperatures' is connected to 'SWR development' but not to 'its complete disappearance'. Changed to occurrence

P20L23: 'change' -> 'changes'.

P21L11: please replace 'lower SWR soil' , for instance, by ' soil with low degree of SWR'.

P21L15: 'DOC' -> 'dissolved organic C'. It should read: 'higher nutrient and DOC concentration'. Please check ' in the water in wetter zones' (do you mean 'more wettable zones'?).

P21L19: What is meant by C fluxes here? Do you mean other C fluxes than CO2 evolution? If yes, then they should be

specified. Otherwise 'C fluxes' should be deleted.

P22L6: 'were -> 'was'.

P22L15: 'were -> 'was'. Please add a blank after SWC.

P22L19: 'soils' -> 'sites'.

P22L22: 'C' -> 'SOM'.

P23L2-4: This sentence is difficult to understand. Could you please rephrase it.

P23L4-7: Please check the consistency of this sentence ('... include soil wettability with ... high (extreme) SWR?)

P23L14: it should read '.. when soil temperature was high and SWC was low ...'.

P23L25: please add 'climate' after 'temperature'.

P24L7-8: please rephrase: 'Water filled pores are expected to have ... concentrated nutrients'.

P24L13: please insert 'is' after 'this'.

P24L19: 'below' -> 'above'.

P24L21: why differentiating between particle and pore surfaces? These are two sides of the same coin.

P24L24: 'model' -> 'figure'. 'Fig. 9' -> 'Fig. 6'.

P27L3: 'leads' -> 'lead'.

P27L4: 'dominated' -> 'associated'? the word dominated is correct, for clarification the sentence has been rewritten

P27L11: 'results in' -> 'is associated with'.

P27L21: please add 'also' to read 'is also of'.

P27L22: please add 'a better' to read 'then allow a better prediction'

[revised manuscript text omitted]